METHODS AND RESOURCES

# Multi-tissue characterization of the constitutive heterochromatin proteome in *Drosophila* identifies a link between satellite DNA organization and transposon repression

**Ankita Chavan**[1,2,3☉], **Lena Skrutl**[1,2☉], **Federico Uliana**[1,3¤], **Melanie Pfister**[1], **Franziska Brändle**[1,2], **Laszlo Tirian**[4], **Delora Baptista**[5], **Dominik Handler**[4], **David Burke**[6], **Anna Sintsova**[7], **Pedro Beltrao**[8], **Julius Brennecke**[4], **Madhav Jagannathan**[1,3]*

**1** Institute of Biochemistry, ETH Zürich, Zürich, Switzerland, **2** Life Sciences Zürich Graduate School, Zürich, Switzerland, **3** Bringing Materials to Life Consortium, Zürich, Switzerland, **4** Institute of Molecular Biotechnology of the Austrian Academy of Sciences, Vienna BioCenter, Vienna, Austria, **5** Instituto Gulbenkian de Ciência, Oeiras, Portugal, **6** European Molecular Biology Laboratory, European Bioinformatics Institute (EMBL-EBI), Cambridge, United Kingdom, **7** Institute of Microbiology, ETH Zürich, Zürich, Switzerland, **8** Institute of Molecular Systems Biology, ETH Zürich, Zürich, Switzerland

☉ These authors contributed equally to this work.
¤ Current address: Johannes Gutenberg University Mainz, Mainz, Germany
* madhav.jagannathan@bc.biol.ethz.ch

## Abstract

Noncoding satellite DNA repeats are abundant at the pericentromeric heterochromatin of eukaryotic chromosomes. During interphase, sequence-specific DNA-binding proteins cluster these repeats from multiple chromosomes into nuclear foci known as chromocenters. Despite the pivotal role of chromocenters in cellular processes like genome encapsulation and gene repression, the associated proteins remain incompletely characterized. Here, we use 2 satellite DNA-binding proteins, D1 and Prod, as baits to characterize the chromocenter-associated proteome in *Drosophila* embryos, ovaries, and testes through quantitative mass spectrometry. We identify D1- and Prod-associated proteins, including known heterochromatin proteins as well as proteins previously unlinked to satellite DNA or chromocenters, thereby laying the foundation for a comprehensive understanding of cellular functions enabled by satellite DNA repeats and their associated proteins. Interestingly, we find that multiple components of the transposon-silencing piRNA pathway are associated with D1 and Prod in embryos. Using genetics, transcriptomics, and small RNA profiling, we show that flies lacking D1 during embryogenesis exhibit transposon expression and gonadal atrophy as adults. We further demonstrate that this gonadal atrophy can be rescued by mutating the checkpoint kinase, *Chk2*, which mediates germ cell arrest in response to transposon mobilization. Thus, we reveal that a satellite DNA-binding protein functions during embryogenesis to silence transposons, in a manner that is heritable across later stages of development.

**Data Availability Statement:** The mass spectrometry data have been deposited to the ProteomeXchange Consortium via the PRIDE partner repository with the following dataset identifiers: PXD044367 (IP-MS of D1 from Testis), PXD043237 (IP-MS of Piwi from Embryo), PXD043236 (IP-MS of Prod and D1 in Ovary) and PXD043234 (IP-MS of Prod and D1 in Embryo). Small RNA sequencing and RNA sequencing data have been deposited to GEO under the accession number, GSE283701. Source data for all figures can be found in S1–S9 Tables and S1–S7 Data files. Raw images for Western blots and co-IP can be found in S1 Raw Images. We have also generated a web app to facilitate browsing of the proteomics data (https://d1-prod-interactors-bcfffefc8a0d.herokuapp.com/).

**Funding:** This work was supported by the Swiss National Science Foundation (Project grant 310030_189131 to MJ). The funder had no role in study design, data collection and analysis, decision to publish, or preparation of the manuscript.

**Competing interests:** The authors have declared that no competing interests exist.

**Abbreviations:** AFM, AlphaFold Multimer; BSA, bovine serum albumin; DSB, double-strand break; GSC, germline stem cell; Hmr, hybrid male rescue; IDR, intrinsically disordered region; ipTM, interface predicted template modelling; Lhr, lethal hybrid rescue; PFA, paraformaldehyde; PIC, protease inhibitor cocktail; piRNA, Piwi-interacting RNA; PTM, posttranslational modification; ROI, region of interest; Sov, small ovary; TE, transposable element.

## Introduction

"Heterochromatin" is classically defined as regions on eukaryotic chromosomes that remain condensed throughout the cell cycle [1]. The sequences that constitute heterochromatin mainly consist of simple noncoding tandem repeats known as satellite DNA [2] and interspersed repeats known as transposable elements (TEs) [3]. Both types of repeats can account for significant fractions of eukaryotic genomes and previous studies have suggested that they may propagate selfishly at the expense of overall organismal fitness. In addition, unrestricted expression of satellite DNA repeats and TEs can lead to adverse outcomes such as genome instability, chromosome mis-segregation, and cell death [4–7]. However, accumulating evidence suggests important functions for regions enriched in these repetitive sequences. For example, satellite DNA repeats at the centromere have been proposed to function in the stable propagation of centromere identity [8,9] while pericentromeric satellite DNA repeats are thought to be important for chromosome pairing [10,11] and chromosome segregation [12,13]. Similarly, regulated activity of TEs can benefit cellular function [14] through roles in telomere maintenance [15], repeat copy number stability [16,17], genome restructuring [18], and the birth of new genes.

Within the nucleus, constitutive heterochromatin is organized into distinct domains that are separate from the more transcriptionally active euchromatin. For example, pericentromeric satellite DNA repeats from multiple chromosomes are clustered by satellite DNA-binding proteins into DNA-dense nuclear foci known as chromocenters, which typically remain transcriptionally silent [19,20]. This transcriptional silencing is orchestrated, in part, by the deposition of repressive posttranslational modifications (PTMs) at target chromatin loci [19]. Once established, this repressive chromatin state can be heritably propagated, both across cell divisions within an organism as well as across generations [21–24]. Intriguingly, abrogation of heterochromatin-associated PTMs does not always perturb the highly condensed state and transcriptional inactivity of the underlying repetitive sequences [24,25]. In these instances, sequence-specific satellite DNA-binding proteins have been proposed as a complementary mechanism that maintains DNA compaction and transcriptional silencing at constitutive heterochromatin [24,25]. However, experimental evidence as to whether satellite DNA-binding proteins regulate transcriptional silencing at constitutive heterochromatin is lacking.

In *Drosophila*, we have shown that 2 sequence-specific DNA-binding proteins, D1 and Prod, contribute to chromocenter formation by clustering their cognate satellite DNA repeats [20,26,27]. Strikingly, chromocenter disruption through mutation of *D1* or *prod* leads to defective genome encapsulation in the form of micronuclei, which is associated with DNA damage and cell death [26,27]. However, these phenotypes occur in distinct tissues, with male germ cells acutely sensitive to *D1* mutation while larval imaginal discs fail to develop in the absence of Prod [26,27]. These findings point to potential tissue-specific differences in chromocenter composition and regulation. However, the chromocenter-associated proteome in distinct tissues remains largely unidentified since most proteomic studies have typically used constitutive heterochromatin proteins [28–30] or centromeric proteins [29,31,32] as baits in cultured cells.

In this study, we set out to identify chromocenter-associated proteins in multiple *Drosophila* tissues using quantitative mass spectrometry. We used the satellite DNA-binding proteins, D1 and Prod, as baits and identified chromocenter-associated proteins from embryos, adult ovaries, and germline stem cell (GSC)-enriched adult testes. We have identified 196 proteins significantly enriched after affinity purification of D1 and Prod across all tissues. Interestingly, pathway analysis indicated that the D1- and Prod-associated proteome in embryos was enriched in DNA repair and transposon repression proteins. Following up on these data, we have identified an unanticipated molecular relationship between satellite DNA organization

and TE repression in *Drosophila*. Taken together, our study lays the groundwork toward a comprehensive understanding of the multifaceted contributions of satellite DNA repeats and their binding proteins to cellular function.

## Results

### Establishment and validation of strains carrying GFP-tagged satellite DNA-binding proteins

We performed affinity purification coupled to mass spectrometry (AP-MS) to identify chromocenter-associated proteins in *Drosophila* embryos, ovaries, and GSC-enriched testes using D1 and Prod as baits (Fig 1A). In nuclei from these tissues, D1 and Prod foci were often found in proximity to each other (Fig 1B), and we expected to identify common interactors between the 2 baits. To minimize variability between baits, we used an affinity purification strategy involving pulldown of GFP-tagged baits using GFP nanobody-coupled beads. For D1, we made use of an available strain where GFP has been inserted upstream of the endogenous locus. To validate this strain, we assessed the number of chromocenters (D1 and Prod foci per cell) in male and female GSCs as well as fertility in both sexes in comparison to a wild-type reference strain (*D. melanogaster yw*). We observed that the number of D1 and Prod foci per cell in male and female GSCs was identical to the wild type (Fig 1C and 1D), indicating that chromocenter formation was unperturbed in the GFP-D1 strain in these cell types. In addition, we observed that homozygous GFP-D1 males and females were fertile, although the fertility of homozygous GFP-D1 females is reduced compared to the wild type (Fig 1E and 1F). For Prod, we first generated a transgenic strain where a GFP-tagged codon-optimized *Prod* was placed downstream of a 400 bp promoter sequence and inserted into the attP40 locus (S1A Fig). The *GFP-prod* transgene fully rescued the larval lethality associated with loss-of-function *prod* mutations, *prod^{k01180}* and *prod^{U}* [33] (S1B Fig). Hereafter, GFP-Prod refers to the *GFP-prod* transgene in a *prod* mutant background (*p400-GFP-prod*, *prod^{k01180}*/*prod^{U}*) (S1B Fig). We observed that the number of GFP-Prod foci was at wild-type levels in male and female GSCs (Fig 1C and 1D), suggesting that the transgene does not interfere with chromocenter formation in these cells. Like the GFP-D1 strain, GFP-Prod males were fully fertile, while GFP-Prod females were sub-fertile in comparison to the wild type (Fig 1E and 1F). However, ovaries from both the GFP-D1 and GFP-Prod strains were morphologically similar to the control (S1C Fig), suggesting that early stages of oogenesis are likely unaffected in these strains. As a negative control for GFP-D1 and GFP-Prod, we used a strain ubiquitously expressing nucleoplasmic GFP (NLS-GFP). The NLS-GFP did not enrich at chromocenters, and we observed that the number of chromocenters was comparable to the wild type, GFP-D1 and GFP-Prod strains (Fig 1C and 1D). Therefore, the GFP-tagged D1 and Prod strains and the NLS-GFP control strain were used to characterize the chromocenter-associated proteome in intact *Drosophila* tissues.

### Multi-tissue characterization of the chromocenter-associated proteome

First, we optimized lysis and affinity purification conditions such that our bait proteins were effectively solubilized (S1D Fig) and immunoprecipitated (S1E Fig). Following this, we affinity purified GFP-D1, GFP-Prod (and NLS-GFP as a control) from the embryo, ovary, and testis lysates and identified the associated proteins by quantitative mass spectrometry. While we were unable to analyze Prod pulldowns specifically in GSC-enriched testes due to poor sample quality, we identified 196 proteins that were significantly enriched in the D1 and Prod pulldowns in comparison to the control pulldown of NLS-GFP across all tissues ($\log_2$FC>1 and

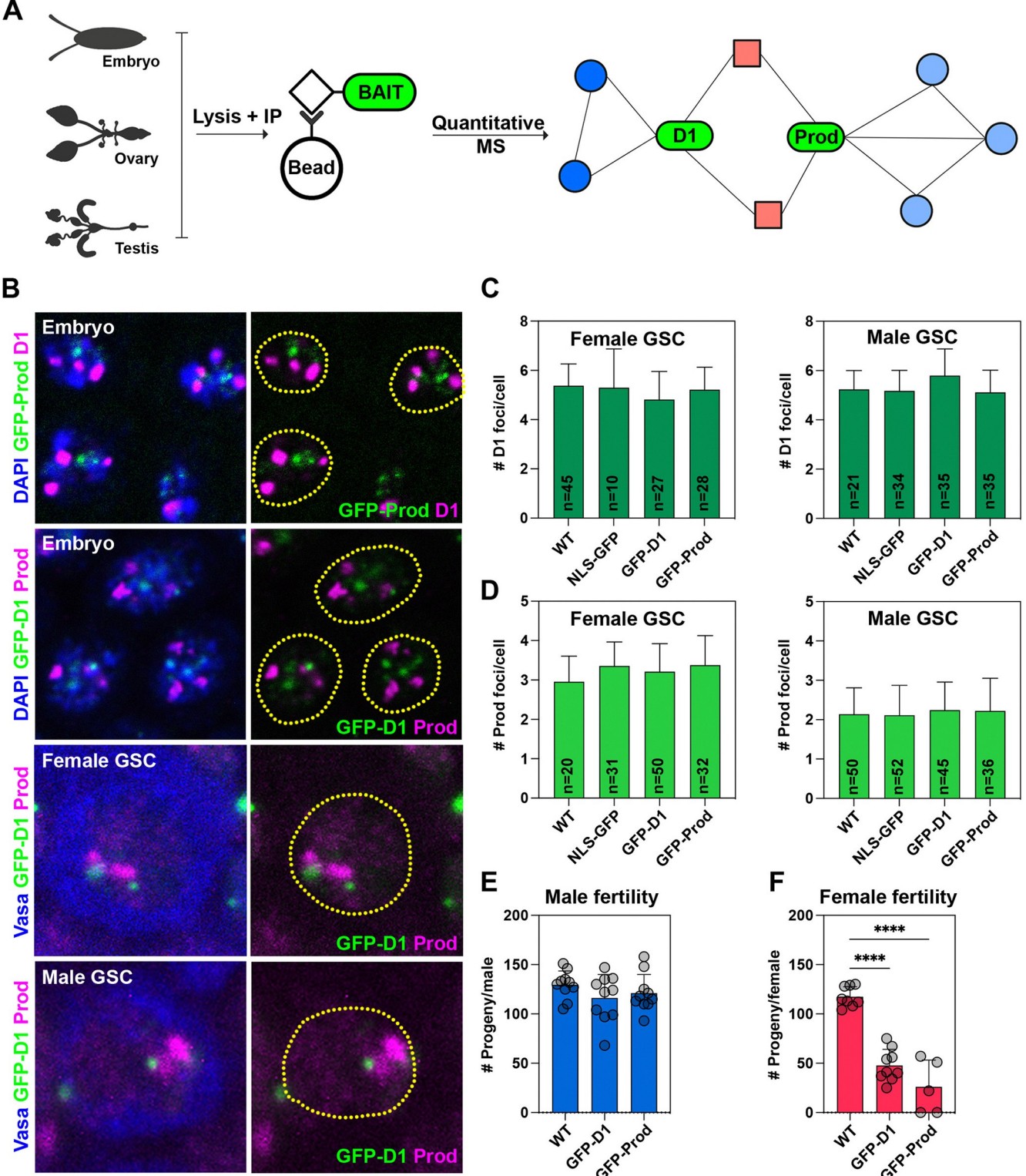

**Fig 1. Experimental setup for the characterization of the chromocenter-associated proteome.** (A) Experimental scheme for the identification of D1- and Prod-associated proteins across multiple *Drosophila* tissues using quantitative mass spectrometry. This figure panel was created using Biorender. (B) D1 and Prod are in proximity within the nucleus in multiple cell types including gastrulating embryos, female germ cells, and male germ cells. (C) Bar plot of the total number of D1 foci present in female GSCs and male GSCs in the wild type (*yw*), *NLS-GFP*, *GFP-D1*, and *GFP-Prod* strains; *n* indicates the number of cells analyzed. (D) Bar plot of the total number of Prod foci present in female GSCs and male GSCs in the wild type (*yw*), *NLS-GFP*, *GFP-D1*, and *GFP-Prod*

strains; *n* indicates the number of cells analyzed. (E, F) Bar plot of the number of progeny sired by a single tester male (E) or female (F) when mated with 2 wild-type flies of the opposite sex over the course of 1 week; **** represents $p < 0.0001$ from an ordinary one-way ANOVA. All error bars are SD and all scale bars are 5 μm. Source data for (C–F) can be found in S1 Data. GSC, germline stem cell.

$p_{adj} < 0.1$, Figs 2A–2C, S2A, S1–S3 Tables), with these hits likely representing both direct and indirect interactions. In embryos, D1 co-purified known chromocenter-associated or pericentromeric heterochromatin-associated proteins, such as Stwl [34–36], ADD1 [28,29], mei-S332 [37], Mes-4 [28], Hmr [29,38,39], Lhr [29,38], and members of the chromosome passenger complex such as Borealin (Borr) and Aurora B [40] (Fig 2A and S1 Table). Prod pulldown in embryos co-purified chromocenter-associated proteins such as Hcs [41] and Saf-B [42] (S1 Table). Our proteomic data also identified some of the previously reported D1/Prod associations such as D1-Xpc (embryo), D1-Stwl (embryo), D1-Smn (ovary), D1-CG5599 (embryo), D1-CG42232 (embryo), Prod-CG15107 (ovary), and Prod-bocks (embryo) [43,44].

We used a less stringent cutoff ($log_2FC > 1$, $p < 0.05$) to analyze our proteomic data across the different baits and tissues, based on the rationale that identical false positive across independent experiments were extremely unlikely. Through this approach, we found that Prod co-purified components of the replication protein A (RPA complex) (S2B Fig), while D1 co-purified multiple proteins involved in DNA replication (S2C Fig) and chromosome condensation (S2D Fig). Interestingly, we observed a substantial overlap between D1- and Prod-associated proteins (yellow points in Fig 2A and 2B and S1–S3 Tables) from the same tissue, with 58 hits pulled down by both baits (blue arrowheads, Fig 2C) in embryos and ovaries. This observation is consistent with the fact that both D1 and Prod occupy sub-domains within the larger constitutive heterochromatin domain in nuclei [27]. Surprisingly, only 9 proteins were co-purified by the same bait (D1 or Prod) across different tissues (magenta arrowheads; Fig 2C and S1–S3 Tables). In addition, only a few proteins such as an uncharacterized DnaJ-like chaperone, CG5504, were associated with both D1 and Prod in embryos and ovaries (Fig 2D). One interpretation of these results is that the protein composition of chromocenters may be tailored to cell- and tissue-specific functions in *Drosophila*. However, more targeted experiments are required to validate whether certain proteins are indeed tissue-specific interactors of D1 and Prod.

Finally, we used AlphaFold Multimer (AFM) to predict interaction interfaces for a subset D1 and Prod interactors ($log_2FC > 2$, $p < 0.01$) from all tissues and potentially distinguish direct versus indirect interactions [45]. AFM-based structural modeling has significantly advanced identification of protein interaction interfaces and has been used across a wide variety of organisms and protein types [46]. Moreover, the AFM-derived interface predicted template modelling (ipTM) score has been used as a standard metric to evaluate the accuracy of the structural models. Based on previous studies [45,47,48], ipTM scores above 0.5 represent a medium confidence model of the interaction interface while ipTM scores above 0.8 represent high confidence models. Despite both D1 and Prod containing large intrinsically disordered regions (IDRs), which hinder the accuracy of structural predictions, we found that 6/78 D1 interactors and 16/79 Prod interactors yielded an ipTM score above 0.5 (S4 Table). Taken together, we have identified several proteins associated with the satellite DNA-binding proteins, D1 and Prod across *Drosophila* tissues and AFM-based structural modelling suggests that some of these hits could be direct interactors of D1 and Prod.

## DNA repair and TE repression proteins are co-purified by D1 and Prod in embryos

Manual inspection of our data revealed that D1 and Prod in embryos pulled down multiple proteins involved in DNA repair, such as POLDIP2 (D1 and Prod), Top2, Xpc, and Elg1 (all

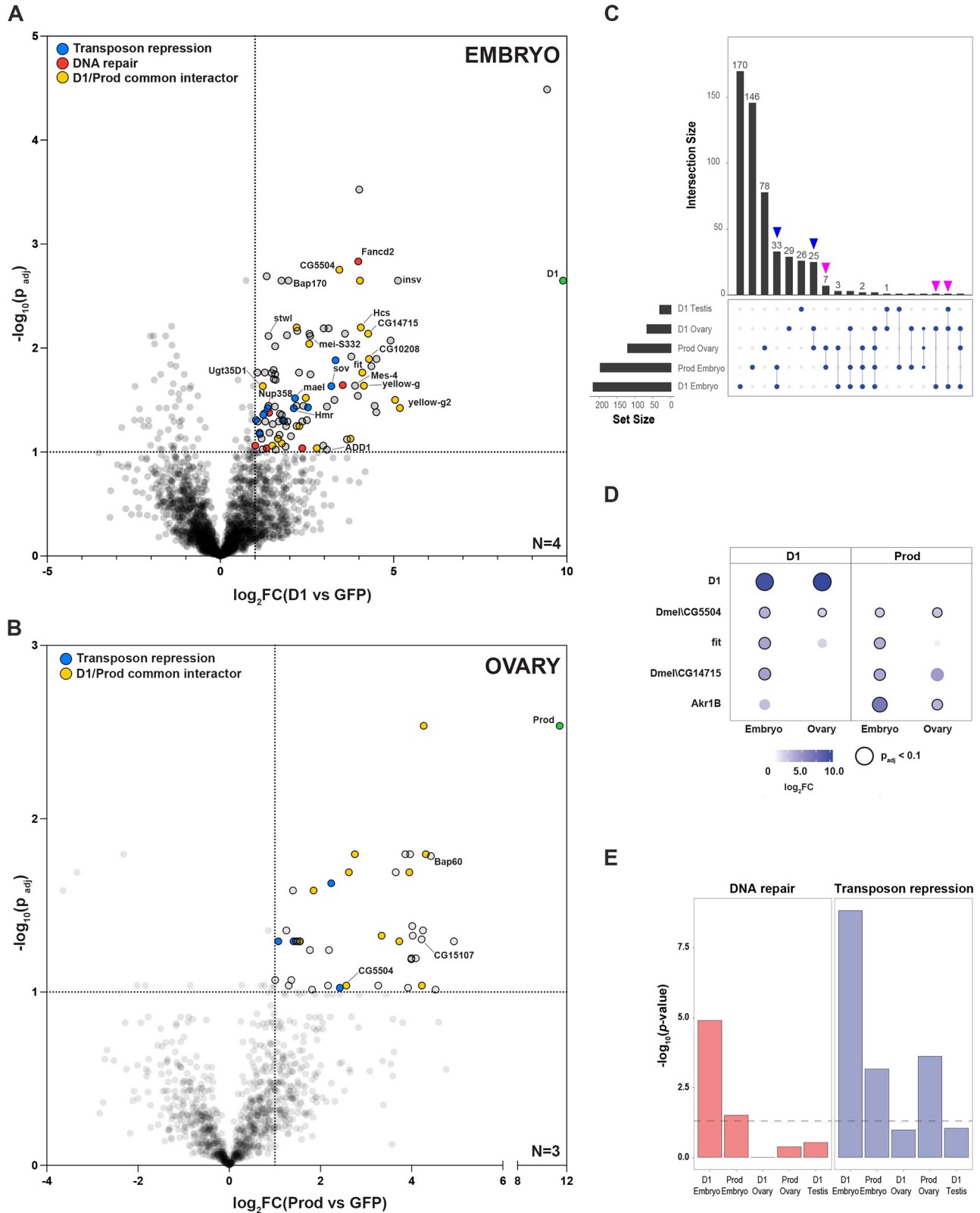

**Fig 2. The chromocenter-associated proteome across *Drosophila* tissues.** (A, B) Volcano plots of the D1-associated proteome in embryos (A) and the Prod-associated proteome in ovaries (B). The dashed lines mark $\log_2FC>1$ and $p_{adj}<0.1$; blue points indicate proteins involved in transposon repression, red points

indicate proteins involved in DNA repair, and yellow points indicate proteins pulled down by D1 and Prod. For proteins belonging to more than one category, we prioritized the representation as follows, transposon repression > DNA repair > D1/Prod common interactors. Number of biological replicates are indicated in the plot. (C) Intersection plot representing number of overlapping proteins in the D1 and Prod pulldowns from different tissues based on a $\log_2$FC>1 and uncorrected $p < 0.05$ cutoff. Blue arrowheads indicate number of proteins common to D1 and Prod pulldown in the same tissue while magenta arrowheads indicate number of common proteins pulled down by D1 or Prod across different tissues. (D) Enrichment of the selected proteins pulled down by D1 and Prod from more than one tissue. The size and color of the circles are proportional to the $\log_2$FC while the black outline indicates $p_{adj}$<0.1. (E) Hypergeometric test of DNA repair or transposon repression pathway proteins from the multi-tissue D1 and Prod interactome based on a $\log_2$FC>1 and uncorrected $p < 0.05$ cutoff. Dashed line indicates $p = 0.05$. Source Data for (A, B, D) can be found in S1 and S2 Tables. Source Data for (C) can be found in S2 Data.

D1 only) (Fig 2A and S5 Table). Tandem repeats like satellite DNA are inherently vulnerable to copy number changes [49] and are known to employ specialized mechanisms during DNA repair [50–54]. Therefore, these interactions may contribute to repeat homeostasis, especially during the rapid early embryonic divisions. In addition, we observed that the canonical TE repression protein, Maelstrom (Mael) was co-purified by D1 and Prod in embryos (Fig 2A and S1 Table). In addition, heterochromatin-associated TE repression proteins such as Sov, Hmr, and Lhr [38,55–57] were also co-purified by D1. Consistently, a hypergeometric test for pathway enrichment revealed that DNA repair and TE repression proteins ($\log_2$FC>1, $p < 0.05$) were enriched in the D1 and Prod pulldowns from embryos and to a lesser extent in other tissues (Fig 2E).

## TE repression proteins are associated with chromocenters

TEs are typically viewed as agents of genomic instability and their unchecked mobilization is usually associated with DNA damage, mutation, cell death, and sterility [58]. In germline tissues from flies, as well as other metazoans, genome defense against transposons requires a dedicated small RNA pathway known as the Piwi-interacting RNA (piRNA) pathway [59,60]. In *Drosophila*, transcription of piRNA precursors is mediated by a protein complex comprised of Rhino, Cutoff, and Deadlock (RDC complex) [61–63], followed by Nxf3-dependent nuclear export of precursor transcripts and processing into mature piRNAs [59,60]. Mature piRNAs are loaded onto a family of RNA-binding Argonaute proteins including Piwi (P-element induced wimpy testis), a canonical member of the piRNA pathway [59,60]. piRNA-bound Piwi enters the nucleus, locates genomic TE loci through sequence complementarity and mediates TE repression through local heterochromatin formation and chromatin compaction. Heterochromatin-associated proteins such as hybrid male rescue (Hmr) [38], lethal hybrid rescue (Lhr) [38], and small ovary (Sov) [55–57] are implicated in heterochromatin formation and transcriptional repression at transposon loci [57]. Of these, Hmr and Sov were co-purified by D1 in embryos in this study. In addition, recent studies in cultured cells of embryonic origin have also identified that both Piwi and Hmr interact with Prod [64]. These data support the notion that satellite DNA-binding proteins may contribute to piRNA pathway function, potentially through regulating heterochromatin stability at TE loci.

In order to validate our proteomics data, we used strains where either Piwi, which is a canonical piRNA pathway Argonaute protein, or Sov, which is implicated in piRNA-dependent heterochromatin formation at TEs [55–57], were tagged with GFP. Although Piwi mediates transcriptional repression at TE loci, it is localized throughout the entire nucleus and the GFP-Piwi signal expectedly overlapped with the heterochromatic D1 and Prod foci in embryonic nuclei (Fig 3A). In contrast, Sov, which is recruited to transposon loci by the piRNA pathway protein Panoramix [57], exhibited a punctate GFP signal in embryonic nuclei, which was frequently present at the boundaries of D1/Prod-containing chromocenters (Fig 3B). We observed a similar pattern for GFP-Piwi and Sov-GFP localization in adult germ cells from males and females (S3A–S3D Fig). Moreover, the D1-interacting heterochromatin protein, Hmr, which contributes to TE repression [38], was also observed at the chromocenter

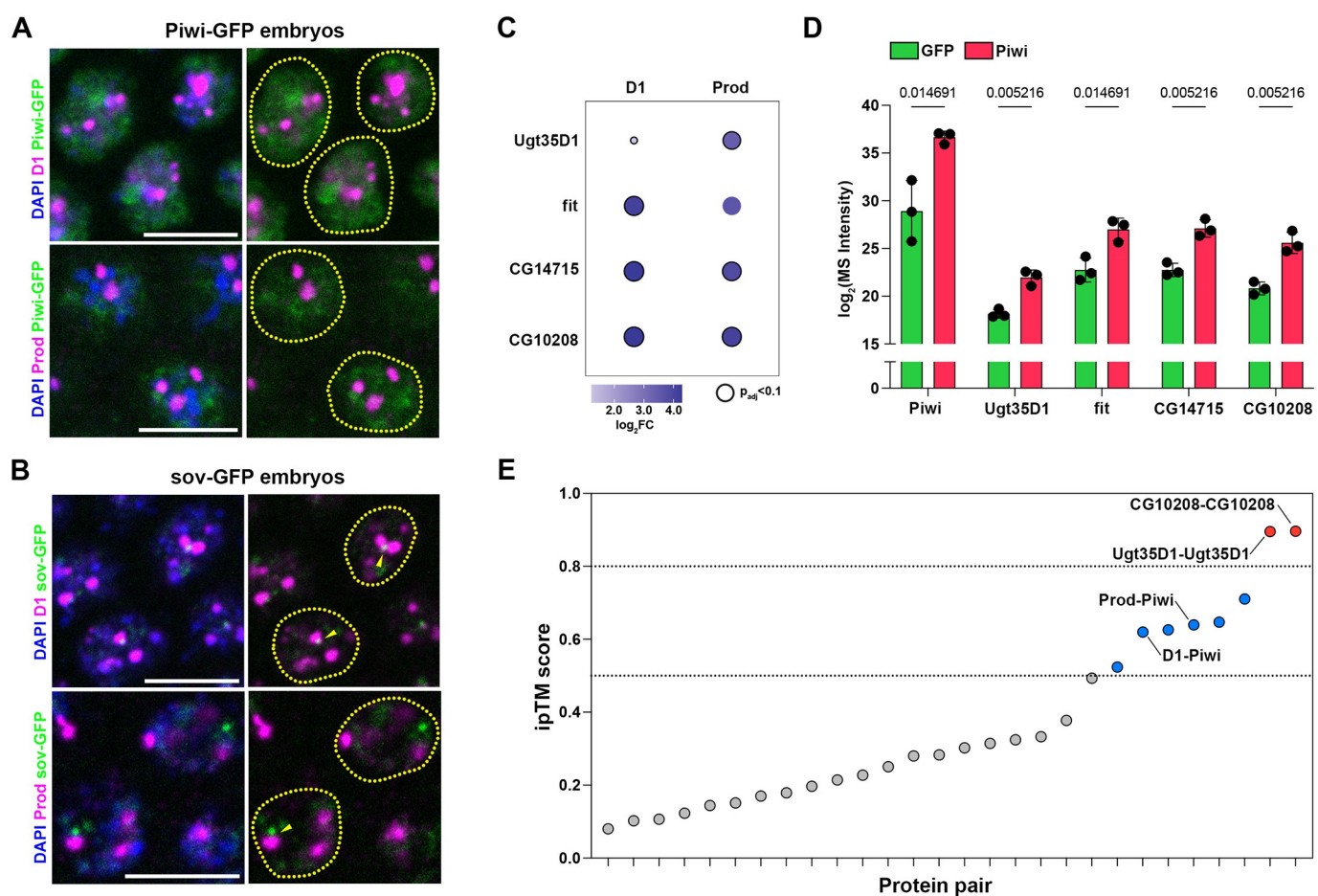

**Fig 3. Molecular relationships between proteins involved in transposon repression and chromocenter formation.** (A, B) GFP-Piwi (A) and Sov-GFP (B) embryos were stained with D1/Prod (magenta) and DAPI (blue). Arrowheads indicate Sov-GFP foci in proximity to chromocenter boundaries. (C) Enrichment of the selected proteins pulled down by D1, Prod, and Piwi from embryos. The size and color of the circles are proportional to the $\log_2$FC while the black outline indicates $p_{adj}<0.1$. (D) Spectral intensities for the indicated proteins from the GFP control (green) and Piwi (magenta) pulldowns from 3 independent replicates. Adjusted $p$-values (Benjamini–Hochberg) are indicated for each comparison. (E) ipTM scores of the pairwise protein structural models based on AFM. Gray points indicate low confidence structural models, blue points indicate medium confidence structural models (ipTM >0.5), and red points indicate high confidence structural models (ipTM >0.8). All scale bars are 5 μm. Source Data for (C, E) can be found in S7 and S8 Tables. AFM, AlphaFold Multimer; ipTM, interface predicted template modelling; Sov, small ovary.

boundary (S3E and S3F Fig), in agreement with recent reports suggesting that Hmr may bridge centromeric and pericentromeric heterochromatin [29,32].

To further test the potential associations between D1/Prod and Piwi, we performed a reciprocal AP-MS experiment using embryonic lysates from the GFP-Piwi strain (and the NLS-GFP strain as a control) (S7 Table). Piwi co-purified proteins identified as D1/Prod interactors in embryonic lysates (Fig 3C and 3D). Strikingly, 4 proteins (CG14715, Fit, CG10208, and Ugt35D1) were high confidence hits from the Piwi, D1, and Prod pulldowns (Fig 3C and 3D). All these proteins are relatively unstudied. CG14715 is an uncharacterized FK506-binding protein, while Ugt35D1 is a putative UDP-glycosyltransferase. Female independent of transformer (Fit) is a sexually dimorphic small peptide that has been implicated in feeding control [65], while CG10208 is an uncharacterized protein with no known domains or function. To determine how these proteins may potentially interact with each other and with D1, Prod, and Piwi, we used AFM to

model interaction interfaces between all possible pairs (S8 Table). Interestingly, 8/28 of the protein pairs had ipTM scores above 0.5 (blue and red points, Fig 3E), with these medium and high confidence scores highlighting interfaces that may give rise to direct interactions. Among these, both D1 and Prod had medium confidence interfaces with Piwi, which were mediated by contacts between disordered regions on D1 and Prod that threaded through a central cleft in the Piwi protein (S4A Fig). At the same time, we were unable to detect a Piwi-D1 interaction by co-immuno-precipitation from embryonic lysates (S4B Fig), suggesting that this interaction may only occur in a small subset of cells (e.g., germ cells) in embryos or that the interaction may be mediated by weak/transient associations between disordered regions, with both possibilities necessitating higher sensitivity approaches like mass spectrometry. Overall, the data from the proteomic, microscopy, and structural modeling experiments strongly hint at a functional relationship between satellite DNA organization and the piRNA pathway.

## Satellite DNA clustering into chromocenters does not rely on piRNA pathway proteins

Repressive histone posttranslational modifications, such as H3K9me2/3, are enriched at pericentromeric heterochromatin [13,25] and recent studies identifying piRNAs complementary to complex satellite DNA repeats [66] suggest that the piRNA pathway may promote repressive PTMs at satellite DNA repeats and facilitate chromocenter formation. This idea was supported by prior studies showing that an interaction between Piwi and the heterochromatin protein, HP1, contributes to heterochromatin organization in both somatic and germ cells [67–69]. Since Piwi mutation results in a severe agametic phenotype in ovaries and testes, we instead used mutations of aubergine, which mediates piRNA amplification through the ping-pong mechanism [60]. In the absence of aubergine, total piRNA levels are decreased and piRNA-free Piwi fails to enter the nucleus [70]. We observed that aubergine mutation ($aub^{HN2}$/$aub^{QC42}$) as well as germline knockdown of aubergine resulted in a mild de-clustering of Prod foci in female GSCs but had no effect on the number of D1 foci in the same cells (S5A–S5C Fig). These findings suggest that the piRNA pathway does not have a prominent role in chromocenter stability in female GSCs.

## Satellite DNA-binding proteins make a subtle contribution to TE repression in adult ovaries

For an initial assessment of whether D1 and Prod contribute to TE repression, we used established reporters of transposon activity in the adult germline (*GFP-Burdock-lacZ*) and adult soma (*gypsy-lacZ*) of *Drosophila* ovaries [71,72]. In the gonadal soma, we used the *tj-gal4* driver to express RNAi against D1 in a genetic background containing the *gypsy-lacZ* reporter. Despite effective depletion of D1 (S6A Fig), we did not observe increased expression of the TE reporter (*gypsy-lacZ*) based on staining for beta-galactosidase (β-Gal) in comparison to the negative control (mCherry RNAi) (S7A Fig). Similar to D1 depletion, Prod depletion did not result in de-repression of *gypsy-lacZ* in the somatic follicle cells (S6B and S7A Figs). In contrast, RNAi-mediated depletion of Sov, which is required for TE repression in gonadal somatic cells[55,56], resulted in β-Gal signal in the follicle cells across multiple stages of oogenesis, indicating expression of the somatic TE reporter (S7A Fig). Next, we tested whether loss of D1 in the germline resulted in increased expression of the germline TE reporter, *GFP-Burdock-lacZ*. To do so, we used a previously validated mutant allele ($D1^{LL03310}$) in trans to a chromosomal deficiency spanning the D1 locus ($Df(3R)^{Bsc666}$) in a background containing *GFP-Burdock-lacZ* [26]. Here, we did not observe GFP signal in the germline cells of both *D1* mutant ovaries and the heterozygous controls (S7B Fig). Similarly, RNAi-mediated Prod depletion

(S6C Fig) in a background containing *GFP-Burdock-lacZ* also did not lead to observable GFP signal in the ovaries (S7B Fig). In contrast, depletion of aubergine, which is required for TE repression in germ cells, resulted in a strong germline GFP signal (S7B Fig). These data suggest that D1 and Prod do not contribute to the repression of *gypsy* and *Burdock* during oogenesis. In addition, previous studies have determined that Prod depletion in adult ovaries does not alter the expression of other TEs such as *HeT-A*, *TAHRE*, and *blood* [71,72]. However, we cannot exclude from these data that D1 and Prod contribute to the repression of other TEs.

Therefore, we performed RNA sequencing to determine transcriptome-wide effects on TE expression. Since Chk2 arrests germ cell development in response to TE de-repression and DNA damage [73,74], we used heterozygous and *D1* mutant ovaries in a *chk2* mutant (*chk2^6006^*) background to prevent developmental arrest of potential TE-expressing germ cells. We observed that both genotypes exhibited similar gonad morphology (Fig 4A). Consistent

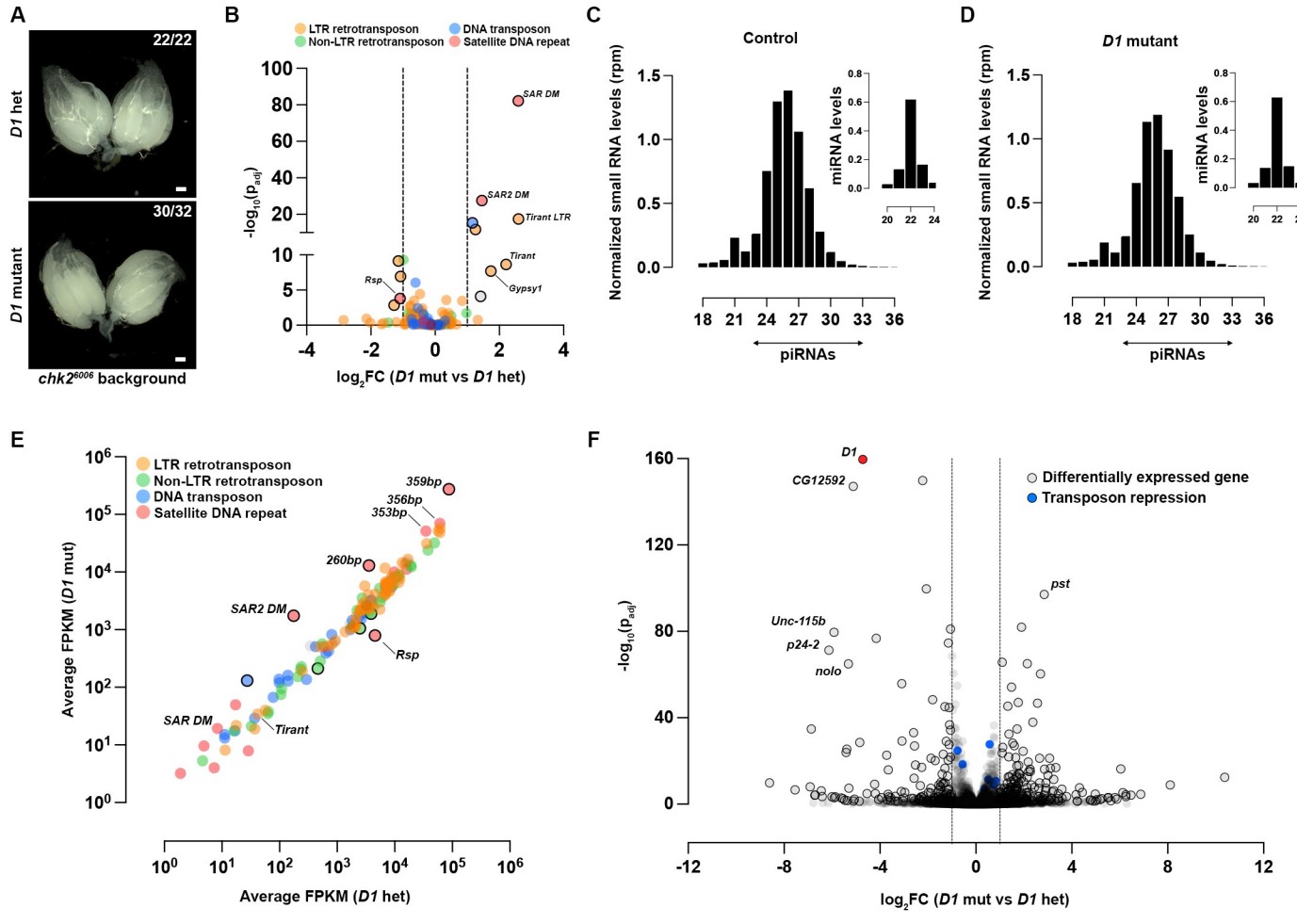

**Fig 4. The effect of D1 on piRNA biogenesis and TE expression in ovaries.** (A) Brightfield images of ovaries from *D1* heterozygous and *D1* mutant females in a *chk2^6006^* background. Scale bars: 100 μm. (B) Volcano plot of differentially expressed transposon families in ovaries from *D1* heterozygous and *D1* mutant females in a *chk2^6006^* background ($n = 3$). Statistically significant changes ($\log_2FC > 1$, $p_{adj} < 0.05$) are indicated with the black outlines. (C, D) Size profiles of Argonaute-bound small RNAs and miRNAs isolated from *D1* heterozygous and *D1* mutant ovaries. Reads from 3 replicates per genotype were normalized per million miRNA reads. (E) Average FPKM for piRNAs mapping antisense to transposon consensus sequences from D1 heterozygous and D1 mutant ovaries ($n = 3$). Black outline indicates statistically significant changes in antisense piRNAs ($\log_2FC > 1$, $p_{adj} < 0.05$). (F) Volcano plot of differentially expressed genes in ovaries from *D1* heterozygous and *D1* mutant females in a *chk2^6006^* background ($n = 3$). Statistically significant changes ($\log_2FC > 1$, $p_{adj} < 0.05$) are indicated with the black outlines. Blue points indicate the 347 genes associated with transposon repression (S6 Table). Source data for (B, F) can be found in S9 Table. Source data for (C–E) can be found in S3 Data. piRNA, Piwi-interacting RNA; TE, transposable element.

with our data from the TE reporters and previous studies, expression of most TE families was not elevated in *D1* mutant ovaries in comparison to the heterozygous control (Fig 4B and S9 Table). One exception was the LTR retrotransposon *tirant*, which has recently invaded *D. melanogaster* populations [75], and was expressed 5- to 6-fold higher in *D1* mutant ovaries. However, the magnitude of TE expression in the *D1* mutant ovaries is not in the range of what has been observed for ovaries that lack canonical piRNA pathway proteins [63,76]. The mild increase in *tirant* expression is therefore unlikely due to compromised piRNA pathway function.

## Levels of TE-targeting piRNAs are largely unperturbed in the absence of D1

Since transposon silencing by the piRNA pathway relies on the production of piRNAs and their loading onto the Piwi, Aub, and Ago3 Argonaute proteins [60], we performed small RNA sequencing to determine whether the Argonaute-bound piRNA pool was altered in *D1* mutant ovaries in comparison to heterozygous control ovaries. Our data show that total levels of mature piRNAs, normalized to miRNA reads, are not substantially affected by the loss of D1 (Fig 4C and 4D). We also did not observe a marked reduction in antisense piRNAs complementary to TEs in *D1* mutant ovaries, including those mapping to *tirant* (Fig 4E). Our data indicate that the biogenesis of TE-mapping piRNAs is unaffected in the absence of D1. Furthermore, our RNA-seq data also demonstrated that none of the 347 genes that are known to contribute to TE silencing (S6 Table) were differentially expressed in *D1* mutant ovaries (blue points; Fig 4F and S9 Table). Altogether, our data suggest that the subtle changes in the expression of a few TEs in *D1* mutant ovaries are unlikely due to compromised piRNA biogenesis or altered expression of piRNA pathway components. Rather, we propose that impaired heterochromatin stability or other piRNA-independent factors [77,78] in the *D1* mutant could be the underlying cause.

## The piRNA biogenesis machinery is mislocalized to satellite DNA in the absence of D1

In contrast to TE transcripts, we observed that transcripts corresponding to the abundant 1.688g/cm$^3$ satellite DNA repeat (denoted as *SAR DM* and *SAR2 DM*; Fig 4B) were up-regulated in *D1* mutant ovaries in comparison to the heterozygous control. Conversely, transcripts mapping to the Responder (Rsp) satellite DNA repeat were down-regulated in the absence of D1 (Fig 4B). These changes in satellite DNA transcription were correlated with changes in piRNAs originating from these repetitive loci (Fig 4E). We further mapped the small RNA reads to the different monomers that make up the 1.688 g/cm$^3$ satellite DNA and found a specific up-regulation of the 260 bp and 359 bp variants (Fig 4E).

Recent studies have demonstrated that the Rhino-Deadlock-Cutoff complex stimulates transcription of complex satellite DNA repeats, which are then processed into mature piRNAs [66,79]. Interestingly, the absence of a ZAD-zinc finger DNA-binding protein known as Kipferl causes the RDC complex to mislocalize to and transcribe the 359 and Rsp satellite DNA repeats at higher levels in ovaries [79]. Based on our observation that *D1* mutant ovaries exhibit up-regulated transcripts and piRNAs corresponding to the 359 bp satellite DNA (Fig 4B and 4E), we tested whether D1 may also regulate the localization of the RDC complex during oogenesis. To do so, we used a strain expressing GFP-tagged Cutoff (EGFP-Cuff) to assess RDC complex localization in ovarian nurse cells, where pericentromeric satellite DNA repeats from all chromosomes are clustered into a single prominent DAPI-dense and D1-enriched chromocenter (arrowhead; Fig 5A). In the EGFP-Cuff strain, we observed that Cuff was

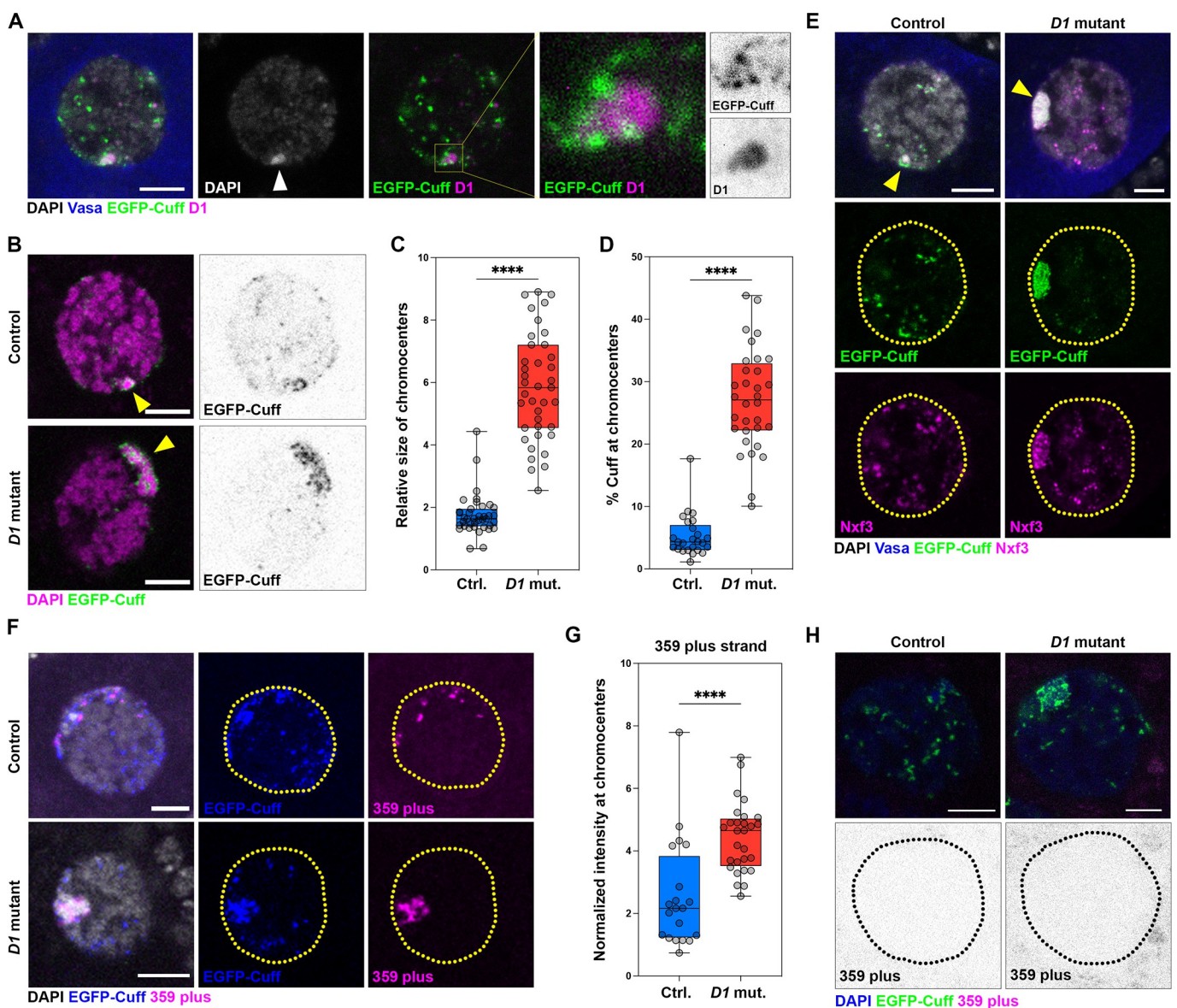

**Fig 5. D1 blocks localization of the piRNA biogenesis machinery to chromocenters.** (A) A single nurse cell from an EGFP-Cuff ovary stained with D1 (magenta), Vasa (blue), and DAPI (gray). The arrowhead points to the DAPI-dense chromocenter. The insets highlight Cuff localization at the periphery of D1-containing chromocenters. (B) Localization of EGFP-Cuff in nurse cell nuclei from control ($D1^{LL03310}$/TM6C) and D1 mutant ($D1^{LL03310}$/$Df(3R)^{Bsc666}$) ovaries. Yellow arrowheads point to the location of the chromocenter. (C) Box-and-whisker plot of the size of nurse cell chromocenters relative to total nuclear size in control ($D1^{LL03310}$/TM6C) and D1 mutant ($D1^{LL03310}$/$Df(3R)^{Bsc666}$) ovaries; **** represents $p < 0.0001$ from a Student's $t$ test. (D) Box-and-whisker plot of EGFP-Cuff enrichment at chromocenters relative to the rest of the nurse cell nucleus in control ($D1^{LL03310}$/TM6C) and D1 mutant ($D1^{LL03310}$/$Df(3R)^{Bsc666}$) ovaries; **** represents $p < 0.0001$ from a Student's $t$ test. (E) Localization of EGFP-Cuff (green) and Nxf3 (magenta) in nurse cell nuclei from control ($D1^{LL03310}$/TM6C) and D1 mutant ($D1^{LL03310}$/$Df(3R)^{Bsc666}$) ovaries. Yellow arrowheads point to the location of the chromocenter. (F) RNA in situ hybridization against the 359 bp satellite DNA transcripts (plus strand, magenta) in control ($D1^{LL03310}$/TM6C) and D1 mutant ($D1^{LL03310}$/$Df(3R)^{Bsc666}$) ovaries in an EGFP-Cuff (blue) background that were also stained for DAPI (gray). (G) Box-and-whisker plot of chromocenter-localized 359 bp satellite DNA transcripts in nurse cells from control ($D1^{LL03310}$/TM6C) and D1 mutant ($D1^{LL03310}$/$Df(3R)^{Bsc666}$) ovaries in an EGFP-Cuff background; **** represents $p < 0.0001$ from a Student's $t$ test. All scale bars are 5 μm. (H) RNA in situ hybridization against the 359 bp satellite DNA transcripts (plus strand, magenta) in control ($D1^{LL03310}$/TM6C) and D1 mutant ($D1^{LL03310}$/$Df(3R)^{Bsc666}$) ovaries in an EGFP-Cuff (green) background that were also stained for DAPI (blue) following RNase A treatment post fixation. Source data for (C, D, G) can be found in S4 Data. piRNA, Piwi-interacting RNA.

localized to the periphery of the nurse cell chromocenter and was also present at many sites across the nucleus, consistent with the presence of piRNA clusters at pericentromeric hetero-chromatin and chromosome arms (Figs 5A, S8A, and S8B). We next examined Cuff localization in *D1* mutant nurse cells in comparison to the heterozygous control. In both the heterozygous control and *D1* mutant, DAPI-dense chromocenters were still detected in nurse cells (yellow arrowheads; Fig 5B). However, we observed an approximately 3-fold increase in chromocenter size in nurse cells lacking D1 (Fig 5B and 5C), likely due to impaired satellite DNA clustering. Strikingly, this increase in chromocenter size in *D1* mutants was accompanied by a substantial mis-localization of Cuff to satellite DNA (Fig 5B and 5D), where we observed approximately 6-fold more Cuff at *D1* mutant chromocenters in comparison to the heterozygous control (Fig 5B and 5D). In addition, we observed that the expanded chromocenters in the *D1* mutant nurse cells were also positive for Nxf3 (Fig 5E), which normally colo-calizes with Cuff at piRNA clusters and mediates nuclear export of piRNA precursors [80,81]. Consistently, RNA in situ hybridization revealed increased transcription of the 359 satellite DNA repeat at the expanded Cuff-positive chromocenters in *D1* mutant nurse cells (Fig 5F and 5G). Finally, the 359 bp probe signal was abolished following RNase A treatment (Fig 5H), which is further indication that the 359 bp satellite DNA repeat is aberrantly transcribed fol-lowing loss of D1. Our data indicate that D1 inhibits RDC localization to certain complex sat-ellite DNA repeats in *Drosophila* ovaries. However, unlike in *Kipferl* mutant ovaries, the RDC mislocalization in *D1* mutant ovaries did not have a substantial effect on the levels of piRNAs targeting TEs and TE silencing. Altogether, the TE reporter experiments, gene expression anal-ysis, and small RNA sequencing suggest that TE repression in the germ cells and somatic cells of adult gonads is largely independent of D1 or Prod.

## Maternal and zygotic D1 contribute to TE repression in adult gonads

Although D1 and Prod appear to be dispensable for TE repression in adult gonads, our proteo-mic data identified that TE repression proteins were mainly co-purified by D1 and Prod in embryos. Thus, testing the functional relevance of these interactions required depleting satel-lite DNA-binding proteins during embryogenesis. This experiment is challenging since $F_1$ mutants (the progeny of heterozygous parents) contain maternal deposits during embryogene-sis (S9A–S9C Fig) and protein/RNA depletion experiments can be ineffective in early embryos [82]. However, the progeny of $F_1$ mutants ($F_2$ mutants) lack both maternal and zygotic protein (S9A–S9C Fig) and allow us to test the role of satellite DNA-binding proteins in TE repression during embryogenesis. Since *D1* mutants are viable and fertile (unlike *prod* mutants, which are larval lethal [33]), we chose to assess TE repression in $F_1$ *D1* mutants, $F_2$ *D1* mutants, and their associated heterozygous controls (S9A–S9C Fig). Briefly, the $F_1$ *D1* mutant lacks zygotically expressed D1 but retains a maternal contribution during embryogenesis and potentially early larval development (S9A and S9C Fig). In comparison, the $F_1$ *D1* heterozygote contains both maternally contributed and zygotically expressed D1 (S9A and S9C Fig). In the $F_2$ *D1* mutants, which are the progeny of $F_1$ *D1* mutant females, both maternal and zygotic D1 are absent dur-ing embryogenesis and adulthood (S9B and S9C Fig), while the corresponding $F_2$ *D1* heterozy-gote lacks maternal D1 in the early embryo but expresses D1 upon zygotic genome activation (S9B and S9C Fig).

We initially assessed female gonad morphology in the above genotypes since unrestricted transposon expression can result in arrested germ cell development and agametic ovaries [4,83]. Strikingly, we observed that $F_2$ *D1* mutant females predominantly exhibited agametic ovaries (Fig 6A), with the vast majority of ovarioles either lacking any discernible Vasa-positive germ cells or containing only undifferentiated Vasa-positive germ cells, which did not develop

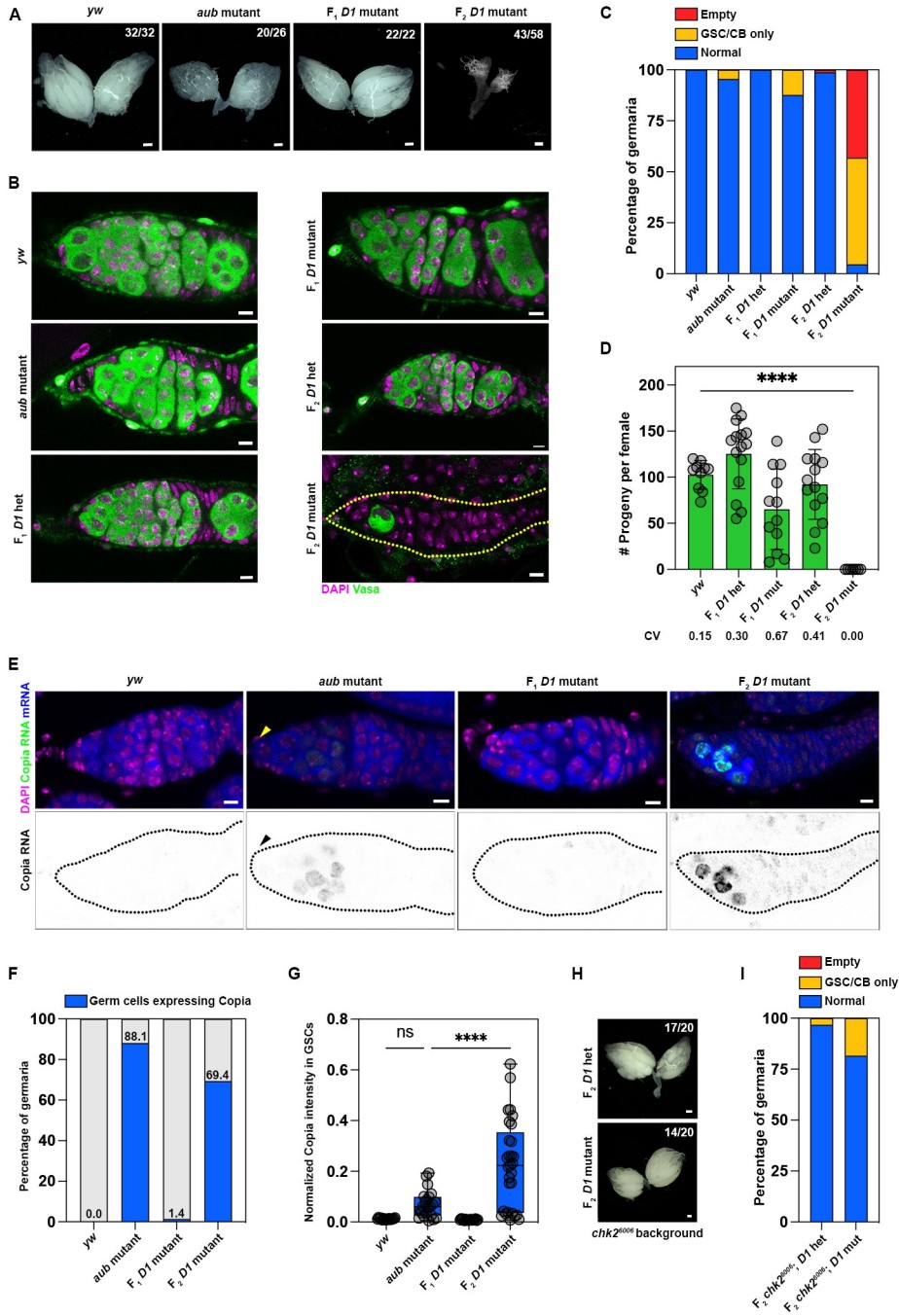

**Fig 6. Loss of D1 during embryogenesis triggers transposon expression in adult ovaries.** (A) Brightfield images of ovaries from the indicated genotypes. Scale bars: 100 μm. (B) Ovaries from the indicated genotypes were stained with Vasa (green) and DAPI (magenta). Scale bars: 5 μm. (C) Percentage of germaria containing a normal population of germ cells, only germline stem cells/cystoblasts (GSC/CB only) or completely lacking germ cells in *yw* (*n* = 48), *aubergine* mutant (*n* = 66), $F_1$ *D1* heterozygous (*n* = 51), $F_1$ *D1* mutant (*n* = 65), $F_2$ *D1* heterozygous (*n* = 81), and $F_2$ *D1* mutant (*n* = 109). (D) Bar plot of the number of progeny sired by a single female of the indicated genotypes when mated with 2 wild-type males over the course of 1 week. CV represents coefficient of variation; **** represents *p* < 0.0001 from an ordinary one-way ANOVA. (E) RNA in situ hybridization against Copia (green) and total polyadenylated mRNA (blue) was performed in ovaries of the indicated genotypes that were also stained for DAPI (magenta). Yellow and black arrowheads indicate undifferentiated germ cells in the *aubergine* mutant germarium. Scale bars: 5 μm. (F) Percentage of germaria with germ cells expressing Copia transcripts in *yw* (*n* = 62), *aubergine*

mutant ($n = 67$), $F_1$ $D1$ mutant ($n = 71$), and $F_2$ $D1$ mutant ($n = 50$). (G) Box-and-whisker plot of copia expression normalized to total polyA mRNA in undifferentiated germ cells (GSCs/CBs) from $yw$ ($n = 16$), *aubergine* mutant ($n = 22$), $F_1$ $D1$ mutant ($n = 18$), and $F_2$ $D1$ mutant ($n = 29$); **** represents $p < 0.0001$ from one-way ANOVA. (H) Brightfield images of ovaries from the indicated genotypes. Scale bars: 100 μm. (I) Percentage of germaria containing a normal population of germ cells, only germline stem cells/cystoblasts (GSC/CB only) or completely lacking germ cells in $F_2$ $D1$ heterozygous ($n = 62$) and $F_2$ $D1$ mutant ($n = 38$) ovaries in a *Chk2* mutant background. Source data for (C, D, G, I) can be found in S5 Data. GSC, germline stem cell.

into cysts (Fig 6B and 6C). Moreover, the failure of germline development and agametic ovaries in the F2 *D1* mutant strongly correlated with loss of female fertility (Fig 6D). Interestingly, the presence of either maternal or zygotic D1 was sufficient to rescue this phenotype, as gonad morphology and early germ cell development of both the $F_1$ *D1* mutant females (maternal D1 only) and the $F_2$ *D1* heterozygote females (zygotic D1 only) were mostly unaffected (Fig 6A–6C), even though both genotypes exhibited more variable fertility (Fig 6D). We next tested whether certain transposons were expressed in $F_2$ *D1* mutant ovaries using single molecule RNA in situ hybridization (smFISH). Here, we observed that the *copia* retrotransposon, but not other TEs such as *burdock* or *gypsy*, was highly expressed in approximately 70% of early germ cells from $F_2$ *D1* mutant ovaries (Fig 6E–6G). In comparison, *copia* expression was not elevated in early germ cells from the *yw* wild-type strain or in the $F_1$ *D1* mutant (Fig 6A). While approximately 90% of the *aubergine* mutant germaria (positive control) contained *copia* expressing germ cells (Fig 6E and 6F), this expression was mostly restricted to differentiated germ cell cysts; undifferentiated *aubergine* mutant germ cells (GSCs and cystoblasts, arrowheads; Fig 6E) exhibited a markedly lower level of *copia* expression (Fig 6G). Thus, the high expression of *copia*, and potentially other TEs, occurring in the early germ cells of $F_2$ *D1* mutants, may contribute to the loss of GSCs or block their differentiation, ultimately leading to agametic gonads. Consistently, we found that *chk2* mutation rescues the agametic gonads in the $F_2$ *D1* mutant (Fig 6H and 6I). Together, our data reveal embryogenesis (or potentially early larval development) as a critical period when the presence of maternal and/or zygotic D1 is sufficient for establishing TE silencing, which can then be heritably maintained throughout development (Fig 7).

## Discussion

In this study, we have characterized the chromocenter-associated proteome across *Drosophila* embryos, ovaries, and GSC-enriched testes using 2 known satellite DNA-binding proteins, D1 and Prod, as baits. We have identified 196 potential interactions across 3 different tissues encompassing known heterochromatin-associated proteins, DNA repair enzymes, proteins involved in transposon repression, and a host of uncharacterized proteins. Interestingly, we found a tissue-specific signature for the chromocenter proteome, with D1 and Prod associating with many common proteins per tissue and relatively few common proteins per bait across the tissues. This finding suggests that the tissue-context or cellular function may be a primary determinant of the chromocenter proteome. In addition, our structural predictions using AFM highlight potential interfaces that may mediate direct interactions between D1, Prod, and their associated proteins. We suggest that these high-scoring hits represent promising avenues of research that will expand our knowledge of satellite DNA biology and chromocenter function.

Since islands of TE are often embedded within vast tracts of satellite DNA [3,17,84], we were intrigued by the hits from the piRNA pathway, which were predominantly identified in embryos. Here, we observed that the absence of D1 during embryogenesis (the $F_2$ *D1* mutant) led to a striking elevation of TE expression and Chk2-dependent tissue atrophy in adult

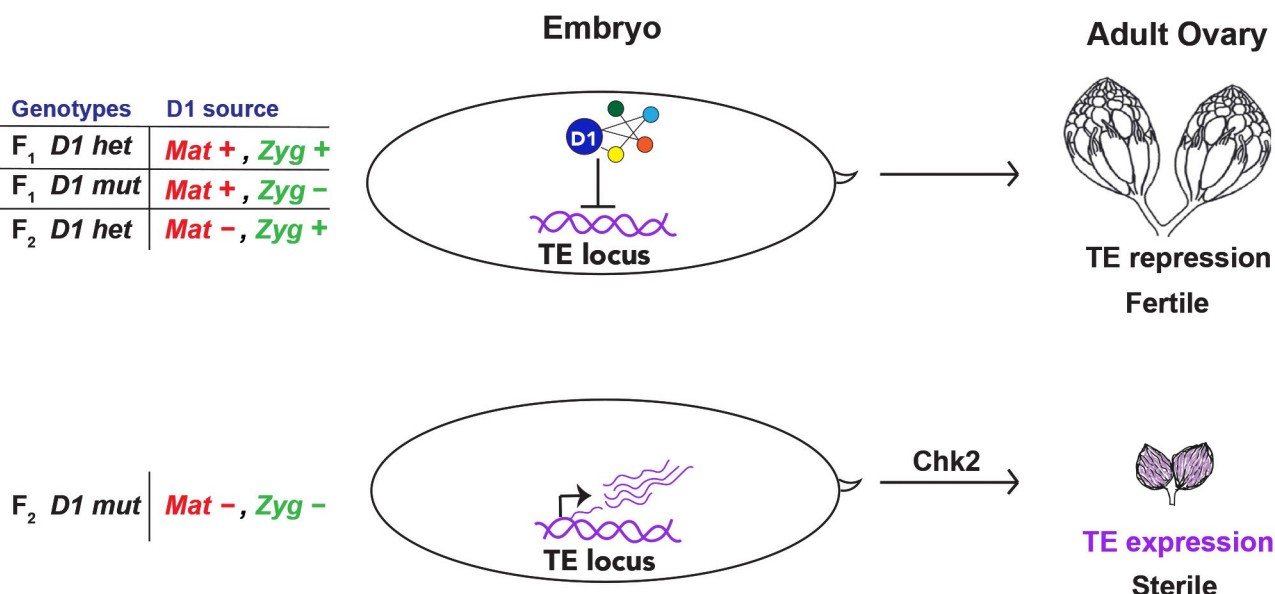

**Fig 7. The role of D1 in heritable TE repression.** An embryonic pool of D1 that originates from either maternal deposition (*Mat* +) and/or zygotic expression (*Zyg* +) is important for TE repression in adult gonads. In the absence of embryonic D1 (F₂ *D1* mt, *Mat–Zyg–*), TE expression contributes to Chk2-mediated gonadal atrophy. Ovary image in the top panel was adapted from [66], which is distributed under the terms of the Creative Commons Attribution License and permits unrestricted use and redistribution provided that the original author and source are credited. TE, transposable element.

ovaries. Similarly, deleterious phenotypes were not observed in adult ovaries, when either only maternally deposited D1 (the F₁ *D1* mutant) or only zygotically expressed D1 (the F₂ *D1* heterozygote) were present during embryogenesis. These data suggest that the presence of D1 throughout development is not per se required for TE repression. Rather, our findings point to embryogenesis as the likely window of time during which D1 contributes to heritable TE silencing in germ cells. They also explain why approaches such as RNAi-mediated knockdown or mutation in the F₁ generation, where D1 likely persists during embryogenesis, do not result in TE expression. Our results also echo previous observations that embryonic Piwi is important for heterochromatin-dependent gene silencing in adults [69], although it should also be noted that the role of embryonic Piwi in transposon silencing is more limited [85,86].

We consider it likely that the TE expression in the F₂ *D1* mutant germ cells results from heterochromatin instability. One straightforward possibility is that loss of heterochromatin at TE loci directly enables transposon expression. However, since piRNA biogenesis relies on heterochromatin modifications such as H3K9me3 and H3K27me3 [87,88], it is also possible that impaired piRNA biogenesis in the F₂ *D1* mutant germ cells may contribute to TE expression and gonadal atrophy. Interestingly, germ cell loss in the F₂ *D1* mutant ovaries was nearly completely rescued by *chk2* mutation. While *chk2*-dependent GSC loss can be triggered by mutations in the piRNA pathway, *chk2* also responds to unresolved DNA double-strand breaks (DSBs) in *Drosophila* female GSCs [89]. Since D1 interacts with a many DNA repair proteins, it is possible that persistent DSBs in F₂ *D1* mutant germ cells, in addition to TE expression, may contribute to *chk2*-dependent germ cell loss.

How could a satellite DNA-binding protein like D1 contribute to heritable silencing at TE loci? While the D1-dependent clustering and compaction of satellite DNA repeats may shield satellite DNA-embedded TE loci from transcription factors, this alone does not explain how TEs remain silenced throughout development in the absence of D1 (e.g., the F₁ *D1* mutant). Rather, we propose that biophysical properties associated with the condensation of satellite DNA repeats

into chromocenters could play an important role. For example, the IDRs present in D1 and Prod [90] could facilitate the selective partitioning of client proteins into chromocenters. Potential client proteins could include piRNA-dependent heterochromatin proteins such as Hmr, Lhr, and Sov, canonical piRNA pathway proteins such as Piwi and Mael or other D1/Prod-interacting heterochromatin proteins identified in this study. In essence, satellite DNA-binding proteins may foster an environment at satellite DNA-embedded TE loci during embryogenesis that facilitates the deposition of repressive PTMs and ensures heritable transcriptional repression.

In summary, our study has characterized the chromocenter-associated proteome across 3 *Drosophila* tissues. We have revealed an unanticipated relationship between satellite DNA organization and heritable transposon repression. The as-of-yet uncharacterized hits identified in this study establish a path toward revealing the full scope of satellite DNA and chromocenter function in *Drosophila* as well as other eukaryotes.

## Materials and methods

### Fly husbandry and strains

All fly stocks were raised on standard Bloomington medium at 25°C unless otherwise indicated. *D. melanogaster yw* was used as a wild-type stock. *prod$^U$* (BDSC42686), *GFP-D1* (BDSC50850), *Ubi-NLS-GFP* (BDSC5826, BDSC5826), *Df(3R)$^{Bsc666}$* (BDSC26518), *aub$^{QC42}$* (BDSC4968), *aub$^{HN2}$* (BDSC8517), *piwi$^1$* (BDSC43637), *piwi$^2$* (BDSC43319), *UAS-Dcr-2* (BDSC24650), *mCherry$^{RNAi}$* (BDSC35785), and *aub$^{RNAi}$* (BDSC39026) were obtained from the Bloomington *Drosophila* stock center. *D1$^{LL03310}$* (DGRC140754) and *FRT42D prod$^{k08810}$* (DGRC111248) were obtained from the Kyoto stock center. *GFP-Burdock-lacZ* (v313212), *tj-gal4, gypsy-lacZ* (v313222), *GFP-Piwi* (v313321), *EGFP-Cuff* (v313269), *D1$^{RNAi}$* (v101946), and *Prod$^{RNAi}$* (v106593) were obtained from the Vienna Drosophila Resource Center. *Hmr-HA* [38] was a gift from Daniel Barbash. Sov-GFP (*sov$^{def1}$*; *Dp(1;2)$^{FF025056}$*) [55] was a gift from Ferenc Jankovics. *nos-gal4* (second chromosome insertion) and *chk2$^{6006}$* were gifts from Yukiko Yamashita. *nos-gal4* (third chromosome insertion) [91] and UAS-Upd [92] have been previously described.

### Construction of the GFP-Prod strain

First, a 400 bp promoter upstream of the Prod start site from *D. melanogaster* was PCR amplified using the following primer pair, GATCAAGCTTCTGTTGTTATGCATATCGTTC and GATCGAATTCCCGGGTATCCTTGCTC and subcloned into the HindIII and EcoRI sites on *pUASt-GFP-attB*, replacing the UAS sequence. Subsequently, a codon-optimized Prod sequence was inserted downstream of the GFP, and transgenic flies were generated using PhiC31 integrase-mediated transgenesis into the *attP40* site (BestGene). The GFP-Prod allele was generated by recombining the *p400-GFP-Prod* transgene with *prod$^{k08810}$*. Two virgin *prod$^U$* females were crossed to single tester male carrying either the *p400-GFP-Prod*, *prod$^{k08810}$* allele, or the *prod$^{k08810}$* allele as a control in vials at 25°C. The percent of trans-heterozygous *prod* mutant flies was quantified in each replicate.

### Fertility assay

For male fertility assays, a single tester male was maintained with 2 virgin *yw* females in vials for 7 days and the number of resulting progenies was counted until 20 days post setup. For female fertility assays, a single tester female was maintained with 2 *yw* males in vials for 7 days and the number of resulting progenies was counted until 20 days post setup. Vials that contained any deceased parent flies were omitted from the analysis.

## Immunofluorescence staining

Immunofluorescence staining of *Drosophila* tissues was performed as follows. For fixation, adult tissues were dissected in PBS, transferred to 4% formaldehyde in PBS, and fixed for 30 min. Fixation of *Drosophila* embryos was performed using a standard protocol. Briefly, collected embryos were moved to glass scintillation vials and washed 3 times with 1× PBS, dechorionated using 3 ml of 50% bleach solution for 90 s by vigorous shaking and washed again in 1× PBS 5 times to remove the bleach. Dechorionated embryos were fixed using a heptane: 4% paraformaldehyde (PFA) (1:1) solution by vigorously shaking for 20 min. Following fixation, the bottom aqueous phase was removed and ice-cold methanol was added to devitellinize the fixed embryos. After this step, embryos were washed 3 times in ice-cold methanol and stored in methanol at −20°C for further use. Prior to immunofluorescence staining, embryos were rehydrated using serial washes of methanol: PBS-T at 4:1, 1:1, and 1:4 volumes and followed by 1 wash in 1× PBS-T for 5 min each. For immunostaining, fixed tissues were washed in 1× PBS-T (PBS containing 0.1% Triton-X) for at least 60 min, followed by incubation with primary antibody in 3% bovine serum albumin (BSA) in PBS-T at 4°C overnight. Samples were washed 3 times in PBS-T, incubated with secondary antibody in 3% BSA in PBS-T at 4°C overnight, washed as above, and mounted in VECTASHIELD with DAPI (Vector Labs). Fluorescent images were taken using a Leica TCS SP8 confocal microscope with 63× oil-immersion objectives (NA = 1.4). Images were processed using Adobe Photoshop or ImageJ.

## RNA in situ hybridization

RNA in situ hybridization on ovaries was performed as previously described [93]. Briefly, 3 to 4 ovaries were dissected in RNase-free 1× PBS and fixed in 3.7% formaldehyde in PBS for 30 min on a nutator with gentle shaking. Following fixation, samples were washed with RNase-free 1× PBS 3 times and subsequently incubated in ice-cold ethanol at 4°C overnight on a nutator. The next day, samples were washed with RNase-free wash buffer (2XSSC, 10% formamide) for 3 min at room temperature and subsequently incubated with the hybridization mix (50 to 125 nM probes, 2× SSC, 10% dextran sulfate, 1 g/L *E.coli* tRNA, 2 mM vanadyl ribonucleoside complex, 0.5% Rnase-free BSA, and 10% deionized formamide) at 37°C overnight. Samples were then washed twice (wash buffer) at 37°C for 30 min each and mounted in VECTASHIELD with DAPI (Vector Labs). The following probe was used to hybridize to the 359 bp satellite plus strand (AGGATTTAGGGAAATTAATTTTTGGATCAATTTTCGCATTTTTTGTAAG). Stellaris RNA FISH probes against the full-length *burdock* and *gypsy* consensus sequences were synthesized (LGC Biosearch technologies). Stellaris RNA FISH probes targeting *Het-A* and *copia* were gifts from Yukiko Yamashita. An oligo-dT probe was used to hybridize to polyadenylated RNA. Fluorescent images were taken using a Leica TCS SP8 confocal microscope with 63× oil-immersion objectives (NA = 1.4). Images were processed using Adobe Photoshop.

## Image analysis

All image analysis was performed using ImageJ. For the quantitation of relative chromocenter size, we demarcated a region of interest (ROI$^N$) encompassing the entire nucleus as well as second ROI encompassing only the chromocenter (ROI$^C$) for each nurse cell (stages 4–6) analyzed. ROI$^C$/ROI$^N$ equals the relative chromocenter size. For enrichment of EGFP-Cuff at chromocenters, we estimated the total EGFP-Cuff signal in ROI$^C$ and divided it by the total EGFP-Cuff signal in ROI$^N$ to give the percentage of EGFP-Cuff at chromocenters. For the normalized estimation of Copia expression, we estimated the average intensity of nuclear Copia (ROI$^{Copia}$) and normalized this value to the 3 cytoplasmic ROIs that estimated the average intensity of total mRNA based on the oligo-dT probe. For the normalized estimation of 359

transcripts at chromocenters, we estimated the average intensity of 359 probe signal at chromocenters (ROI$^C$) for each nurse cell and normalized this value to a cytoplasmic ROI that have background fluorescence.

## Isolation of nuclei from embryos

Flies reared in population cages were allowed to lay eggs on fresh yeast paste for approximately 18 h. Embryos were collected using a set of sieves, washed with 1× PBS and dechorionated using 50% bleach. Next, embryos were moved to a dounce homogenizer containing light nuclear buffer (15 mM Tris-HCl (pH 7.4), 60 mM KCl, 5 mM MgCl$_2$, 15 mM NaCl2, and 1 M sucrose) supplemented with 1× protease inhibitor cocktail (PIC, Roche,11697498001) and 1 mM PMSF. The homogenate was then incubated on ice for 5 min and filtered through 2 layers of Miracloth tissue (Calbiochem). The filtered homogenate was then centrifuged for 20 min at 480 rpm at 4˚C to pellet any debris. The supernatant containing nuclei was transferred into a fresh tube and the nuclei were pelleted at 4,000 g for 30 min at 4˚C. The pellet was then resuspended in 1 ml light nuclear buffer containing 0.1 mM EDTA and the nuclei were again centrifuged at 16,000 g for 5 min before lysis.

## Tissue lysis

Nuclei isolated from embryos were lysed for 90 min at 4˚C with extraction buffer (15 mM Tris-HCl (pH 7.4), 60 mM KCl, 5 mM MgCl$_2$, 450 mM NaCl2, 1% Triton-X [vol/vol], 500 units of benzonase, 1× PIC, and 5 µl/ml PMSF). The resulting lysate was separated from the insoluble chromatin by centrifugation for 20 min at 14,000 g at 4˚C. For each ovary replicate, approximately 200 pairs were dissected from <1-day-old flies, flash frozen with liquid nitrogen and stored at −80˚C prior to lysis; 500 µl of ice-cold lysis buffer (450 mM NaCl, 50 mM Tris-HCl (pH 7.5), 5% glycerol, 1 mM MgCl2, 1 mM EDTA, 0.2% NP40, 0.5 mM DTT, 1× PIC, 5 µl/ml PMSF) was added to frozen ovaries, which were then homogenized with a motorized pestle until complete breakdown of the tissue. After homogenization, samples were incubated with benzonase (Sigma Aldrich, 2 µl/500 µl of lysate) for 90 min at 4˚C on a rotating wheel. Following this, samples were first centrifuged for 3 min at 10,000 g at 4˚C to precipitate the debris from the tissues, homogenized again for 30 s and centrifuged for 5 min at approximately 18,700 g at 4˚C. The supernatant (without the surface lipid layer) was transferred to a new 1.5 ml tube and centrifuged again for 5 min at approximately 18,700 g prior to use. For each testis replicate, approximately 100 pairs of GSC-enriched (nos-gal4 > UAS-Upd) testis tumors were dissected, flash frozen with liquid nitrogen, and stored at −80˚C prior to lysis; 300 µl RIPA lysis buffer [450 mM NaCl, 0.5% NP-40 [vol/vol], 0.5% Sodium deoxycholate [mass/vol], 0.1% SDS, 50 mM Tris (pH 7.4)] containing 1× PIC (Roche,11697498001) and 5 µl/ml PMSF (Sigma-Aldrich, P7626) was added to frozen testes for 10 min. Tissue disruption was performed by applying 20 strokes with the tight pestle in a 1 ml dounce homogenizer, the samples were incubated for another 20 min on ice following which benzonase was added for 1 h at 10˚C. Homogenized lysates were centrifuged at 14,000 g for 20 min at 4˚C prior to use. GFP-Piwi and NLS-GFP embryos were collected as indicated in the previous section, dechorionated, flash frozen with liquid nitrogen, and stored at −80˚C prior to lysis. Embryo lysis for these samples was performed in an identical manner to the ovary lysis above.

## Affinity purification and mass spectrometry

Affinity purification was performed using anti-GFP nanobody coupled magnetic beads (approximately 30 µl slurry) (Chromotek AB_2631358). Briefly, beads were collected using a magnetic rack, washed 3 times using the lysis buffer and incubated with the lysate for 90 min

on a rotating wheel at 4˚C. After the incubation, beads were washed 3 times for 5 min with a wash buffer (450 mM NaCl, 50 mM Tris-HCl (pH 7.5), 1 mM MgCl2, 1 mM EDTA, 0.5 mM DTT) at room temperature. Washed beads containing the protein complexes were subjected to on-beads proteolysis. Samples were transferred into a 10 kDa molecular weight cutoff spin column (Vivacon 500, Sartorious), following the FASP protocol [94]. Beads in solution were dried, denatured (8M Urea), reduced (5 mM TCEP, 30 min, 37˚C) and alkylated (10 mM Iodoacetamide, 30 min, 37˚C). Beads were then washed 3 times with 50 mM ammonium bicarbonate (250 μl). During the buffer exchange, samples were centrifuged at 10,000 g. Subsequently, samples were proteolyzed with 0.5 μg of Trypsin (Promega, sequencing grade) for 16 h at 37˚C. The proteolysis was quenched with 5% formic acid and peptides were subjected to C18 cleanup (BioPureSPN PROTO 300 C18, Nest group), following the manufacturer's procedure. The eluted peptides were then dried using a speedvac and resuspended in 20 μl of 2% acetonitrile and 0.1% formic acid.

### Mass spectrometry data acquisition

The MS data sets were acquired with 2 different instrumental setups. The first setup (A) was utilized for the D1, Prod, and NLS-GFP samples from embryos, ovaries, and testes. The second setup (B) was employed for acquiring the Piwi and NLS-GFP samples from embryos. In setup A, LC-MS/MS acquisition was performed on an Orbitrap QExactive+ mass spectrometer (Thermo Fisher) coupled to an EASY-nLC-1000 liquid chromatography system (Thermo Fisher). Peptides were separated using a reverse phase column (75 μm ID × 400 mm New Objective, in-house packed with ReproSil Gold 120 C18, 1.9 μm, Dr. Maisch GmbH) across a gradient from 3% to 25% in 170 min and from 25% to 40% in 10 min (buffer A: 0.1% (v/v) formic acid; buffer B: 0.1% (v/v) formic acid, 95% (v/v) acetonitrile). The DDA data acquisition mode was set to perform 1 MS1 scan followed by a maximum of 20 scans for the top 20 most intense peptides with MS1 scans (R = 70'000 at 400 m/z, AGC = 1e6 and maximum IT = 100 ms), HCD fragmentation (NCE = 25%), isolation windows (1.8 m/z), and MS2 scans (R = 17'500 at 400 m/z, AGC = 1e5 and maximum IT = 55 ms). A dynamic exclusion of 30s was applied and charge states lower than 2 and higher than 7 were rejected for the isolation. In setup B, LC-MS/MS acquisition was performed on an Orbitrap Exploris480 mass spectrometer (Thermo Fisher) coupled to a Vanquish Neo liquid chromatography system (Thermo Fisher). Peptides were separated using a reverse phase column (75 μm ID × 400 mm New Objective, in-house packed with ReproSil Gold 120 C18, 1.9 μm, Dr. Maisch GmbH) across a linear gradient from 7% to 35% in 120 min (buffer A: 0.1% (v/v) formic acid; buffer B: 0.1% (v/v) formic acid, 95% (v/v) acetonitrile). The DDA data acquisition mode was set to perform 1 MS1 scan followed by a maximum of 20 scans for the top 20 most intense peptides with MS1 scans (R = 60'000 at 400 m/z, AGC = 1e6 and auto maximum IT), HCD fragmentation (NCE = 28%), isolation windows (1.4 m/z), and MS2 scans (R = 15'000 at 400 m/z, AGC = 2e5 and auto maximum IT). A dynamic exclusion of 25s was applied and charge states lower than 2 and higher than 6 were rejected for the isolation.

### Mass spectrometry data analysis

Acquired spectra were analyzed using the MaxQuant software package version 1.5.2.8 embedded with the Andromeda search engine against *Drosophila* proteome reference data set (http://www.uniprot.org/, downloaded on 18.01.2021, 22'044 proteins including not reviewed proteins) extended with reverse decoy sequences. The search parameters were set to include only full tryptic peptides, maximum 2 missed cleavage, carbamidomethyl as static peptide modification, oxidation (M) and deamidation (N-ter) as variable modification and "match between

runs" option. The MS and MS/MS mass tolerance was set to 10 ppm. False discovery rate of <1% was used for protein identification. Protein abundance was determined from the intensity of the top 2 unique peptides and were only considered for downstream analysis if identified in all D1/Prod/Piwi replicates. Log2 intensity values were median normalized and missing values were imputed as follows. In instances where the protein of interest was not identified in all control replicates, we imputed the missing values from a normal distribution generated with an average that is 4 times lower than the mean of the D1/Prod/Piwi sample values and a matching standard deviation. In instances where the protein of interest was identified in only 1 control replicate, we imputed the missing values based on the unique value identified for the control and a standard deviation matching the pulldown samples. In instances with only 1 missing value for the control replicates, we imputed using a normal distribution generated with the average and standard deviation of the control values.

Statistical analysis was performed using unpaired two-sided $t$ test and multiple testing correction was performed using the Benjamini–Hochberg method. Hits identified from the differential analysis between the bait pulldown versus the NLS-GFP control, with $log_2FC>1$ and $p_{adj}<0.1$, were considered as interacting proteins. For cross-bait and cross-tissue comparisons in Figs 2C, 2E, and S2B–S2D and structural modeling in S4 Table, we used a relaxed cutoff with $log_2FC>1$ and uncorrected $p < 0.05$. In Prod pulldowns from embryos, we observe both previously identified interactions and heterochromatin-associated proteins suggesting that the surprising absence of detectable Prod spectra is a technical issue. For pathway enrichment analysis, we first generated a list of *Drosophila* DNA repair proteins (GO:0006281, S5 Table) and a custom list of transposon repression proteins based on the literature (S6 Table) and used a hypergeometric test to calculate the enrichment of proteins belonging to these lists in the D1/Prod pulldowns.

## Structural predictions using Alphafold2

To predict the structures of binary protein–protein interactions, we used version 2.3.2 of Alphafold2, which contains updated parameters for multimeric protein–protein interactions [45]. Default values for all parameters and for generating multiple sequence alignments were used. To rank the models according to a confidence metric, we used the ipTM score, which has been show to distinguish interacting from non-interacting proteins [95]. Structural models are available upon request.

## Co-immunoprecipitation and western blotting

For the co-immunoprecipitation, 125 to 250 mg of frozen embryos were thawed on ice and lysed in an identical manner to the ovary lysis protocol detailed above. Approximately 50 μg of lysate was collected as input while approximately 970 μg of lysate at a concentration of 2 mg/ml was incubated with anti-GFP nanobody coupled beads (approximately 50 μl slurry) (Chromotek AB_2631358) for 90 min at 4°C. After the incubation, beads were washed 3 times for 5 min with a wash buffer (450 mM NaCl, 50 mM Tris-HCl (pH 7.5), 1 mM MgCl2, 1 mM EDTA, 0.5 mM DTT) at room temperature. Protein complexes bound to the beads were eluted in 80 μl of SDS sample buffer by heating at 95°C for 5 min. Input samples were also brought to a final volume of 80 μl by adding 30 μl of SDS sample buffer. Equal volumes of input and IP samples were loaded on a 4% to 12% SDS gradient gel (Sigma Aldrich). Immunoprecipitated proteins were detected using mouse anti-GFP (1:1,000, Roche 1181 4460001), guinea pig anti-D1 (1:1,000, [26]), and rabbit anti-histone H3 (1:1,000, Abcam ab1791). Secondary antibodies used were Alexa Fluor Plus 800 donkey anti-mouse (1:10,000, Invitrogen) and donkey anti-guinea pig IR dye 680 (1:10,000, Li-Cor). Western blots were imaged using the Odyssey DLx fluorescence imager (Li-Cor).

## RNA sequencing and data analysis

For each sample, approximately 5 pairs of ovaries from 4-day-old females were dissected in RNase-free 1× PBS and flash frozen in liquid nitrogen until RNA extraction. RNA extraction for each replicate was performed using the RNeasy RNA extraction kit (Qiagen). Samples were treated with DNase post RNA extraction and purified using an RNA purification kit (Promega). RNA concentrations were assessed using a Nanodrop as well as a Qubit RNA analyzer for sample quality and RIN scores. rRNA-depleted libraries were prepared and sequenced by Novogene on an NovaSeq 6000 (Illumina) using paired-end 150 bp sequencing. The resulting raw reads were cleaned by removing adaptor sequences, low-quality-end trimming, and removal of low-quality reads using BBTools v 38.18 (Bushnell, B. *BBMap*. Available from: https://sourceforge.net/projects/bbmap/). The exact commands used for quality control can be found on the Methods in Microbiomics webpage (Sunagawa, S. *Data Preprocessing—Methods in Microbiomics 0.0.1 documentation*. Available from: https://methods-in-microbiomics. readthedocs.io/en/latest/preprocessing/preprocessing.html). The quality-controlled reads were aligned against BDGP6.32 using STAR aligner [96]. Transcript and transposon abundances were quantified using Tetranscripts [97]. Differential gene expression analysis was performed using Bioconductor R package DESeq2 v 1.37.4 [98].

## Small RNA sequencing and data analysis

Small RNA sequencing was performed as previously described [79]. In brief, approximately 5 pairs of ovaries from 3 biological replicates per genotype were lysed and Argonaute-sRNA complexes were isolated using TraPR ion exchange spin columns. sRNAs were subsequently purified using acidic phenol; 3′ adaptors containing 6 random nucleotides plus a 5 nt barcode on their 5′ end and 5′ adaptors containing 4 random nucleotides at their 3′ end were subsequently ligated to the small RNAs before reverse transcription, PCR amplification, and sequencing on an Illumina NovaSeqSP instrument. Raw reads were trimmed for linker sequences, barcodes, and the 4/6 random nucleotides before mapping to the *D. melanogaster* genome (dm6), using Bowtie (version.1.3.0, settings: -f -v 3 -a—best—strata—sam) with 0 mismatches allowed. Genome-mapping reads were crossreferenced with Flybase genome annotations (r6.40) using BEDTools to allow the removal of reads mapping to rRNA, tRNA, snRNA, snoRNA loci, and the mitochondrial genome. For TE mappings, all genome mappers were used allowing no mismatches. Reads mapping to multiple elements were assigned to the best match. Reads mapping equally well to multiple positions were randomly distributed. Libraries were normalized to 1 million sequenced microRNA reads. For calculation of piRNAs mapping to TEs, only antisense piRNAs were considered, and counts were normalized to TE length.

## Supporting information

**S1 Fig. Schematic of the GFP-Prod allele and optimization of sample lysis and immunoprecipitation.** (A) A 400bp promoter sequence upstream of the Prod translation start site was placed upstream of a GFP-tagged and codon-optimized Prod sequence, inserted into the attP40 locus and recombined with the *prod*[k08810] loss-of-function allele. (B) Rescue of prod mutant (*prod*[k08810/U]) lethality was assessed using p400-GFP-Prod, *prod*[k08810] (p400-GFP-Prod, 447 flies were counted from 5 crosses) and *prod*[k08810] (Ctrl, 314 flies were counted from 5 crosses). Red line indicates median while dashed line indicates percent progeny for complete rescue. (C) Brightfield images of ovaries from the indicated genotypes. Scale bars: 100 μm. (D) Western blot of equivalent fractions of the soluble supernatant and insoluble pellet from GFP-Piwi embryos using mouse anti-GFP, guinea pig anti-D1, and rabbit anti-H3 (loading control). (E) Immunoprecipitation of GFP from embryos expressing GFP-Prod or

GFP-D1. Immunoprecipitated proteins were blotted using mouse anti-GFP and are indicated by arrowheads. Expected sizes: GFP-Prod: 75 kDa, GFP-D1: 80 kDa. Source data for (B) can be found in S6 Data. Raw images for (D, E) can be found in S1 Raw Images.
(TIF)

**S2 Fig. Characterization of D1- and Prod-associated proteins in GSC-enriched testes and embryos.** (A) Volcano plot of the D1-associated proteome in GSC-enriched testes. The dashed lines mark $\log_2 FC > 1$ and $p_{adj} < 0.1$. Number of biological replicates are indicated in the plot. (B–D) k-means clustering of embryo hits using STRING-DB show relationships between Prod-associated (B) and D1-associated (C, D) proteins. Source data for (A) can be found in S3 Table.
(TIF)

**S3 Fig. Localization of D1 and Prod with respect to Piwi, Sov, and Hmr in nurse cells and spermatogonia.** GFP-Piwi (A, B), sov-GFP (C, D), and Hmr-HA (E, F) ovaries and testes were stained with D1/Prod (magenta), Vasa/Lamin (blue), and DAPI (gray). Arrowheads indicate sov or Hmr foci in proximity to chromocenter boundaries in individual nurse cells or spermatogonia. Scale bars: 5 μm.
(TIF)

**S4 Fig. Assessing the interactions between Piwi and satellite DNA-binding proteins by AFM structural modelling and co-immunoprecipitation.** (A) Representation of the D1-Piwi and D1-Prod interaction interfaces predicted by Alphafold Multimer structural modelling. The following regions on D1 (120–143, 157–175, and 322–355) and on Prod (120–181 and 320–343) are depicted in magenta. (B) Immunoprecipitation of GFP and Piwi using GFP-trap beads from NLS-GFP (control) and GFP-Piwi embryos. Immunoprecipitated proteins were blotted using mouse anti-GFP and guinea pig anti-D1. Raw images for (B) can be found in S1 Raw Images.
(TIF)

**S5 Fig. Loss of *aubergine* has a mild effect on Prod foci in female GSCs.** (A) Female GSCs expressing *mCherry^RNAi* (*mCherry^GLKD*) or *aub^RNAi* (*aub^GLKD*) under the control of *nos-gal4* stained for Prod (magenta), Lamin (green), and Vasa (blue). Scale bars: 5 μm. (B, C) Box-and-whisker plot of the total number of Prod foci (B) or D1 foci (C) present per female GSC in *mCherry^GLKD* (n = 58 for Prod, n = 60 for D1), *aub^GLKD* (n = 42 for Prod, n = 36 for D1), *aubergine* heterozygous control (Ctrl., n = 38 for Prod, n = 59 for D1) and *aubergine* mutant (*aub^QC42/HN2*, n = 52 for Prod, n = 56 for D1). n indicates the number of cells analyzed; ** indicates $p < 0.01$ based on a one-way ANOVA. Source data for (B, C) can be found in S7 Data.
(TIF)

**S6 Fig. Validation of D1 and Prod knockdowns in gonadal somatic cells and germline cells.** (A, B) Individual egg chambers stained for DAPI (magenta) and Prod/D1 following *tj-gal4* mediated expression of *D1^RNAi* (KK110307) or *Prod^RNAi* (KK104361). *mCherry^RNAi* was used as a control. (C, D) Anterior region of individual ovarioles were stained for DAPI (blue) and Prod (magenta) following *nos-gal4* mediated expression of *mCherry^RNAi* (C) or *Prod^RNAi* (D). Scale bars: 5 μm.
(TIF)

**S7 Fig. Loss of D1 or Prod does not trigger expression of transposon reporters in gonadal somatic cells or germ cells.** (A) Individual egg chambers stained for beta-galactosidase (indicating expression of *gypsy-lacZ*) following *tj-gal4* mediated expression of *mCherry^RNAi*, *sov^RNAi* (HMC04875), *D1^RNAi* (KK110307), or *Prod^RNAi* (KK104361). Arrowhead indicates

follicle cells expressing *gypsy-lacZ*. (B) Anterior region of individual ovarioles were stained for DAPI in the following strains carrying GFP-Burdock-lacZ: $D1^{LL03310}/TM6B$ (Ctrl.) and $D1^{LL03310}/Df(3R)^{Bsc666}$ (*D1* mutant), $nos > mCherry^{RNAi}$ (*mCherry* $^{GLKD}$), $nos > Prod^{RNAi}$ (*Prod* $^{GLKD}$), $nos > aub^{RNAi}$ (HMS01945, *aub* $^{GLKD}$). Arrowheads indicate germ cells expressing GFP-Burdock. Scale bars: 5 μm.
(TIF)

**S8 Fig. Localization of EGFP-Cuff relative to chromocenters.** (A, B) EGFP-Cuff ovaries and testes were stained for D1/Prod (magenta), Vasa/Lamin (blue), and DAPI (gray). Individual nurse cells or spermatogonia are depicted. Scale bars: 5 μm.
(TIF)

**S9 Fig. Putative expression levels of maternal and zygotic D1 through development.** (A, B) Crossing schemes used to generate $F_1$ *D1* mutants and $F_2$ *D1* mutants. (C) Theoretical depiction of maternal deposition and zygotic expression of D1 across embryonic, larval, and adult stages in the indicated genotypes.
(TIF)

**S1 Table. List of proteins detected by LC-MS/MS in embryonic lysates from GFP-D1, GFP-Prod, and NLS-GFP strains.** Fold change of average MS1 intensity and adjusted *p*-value for the GFP-D1 and GFP-Prod strains relative to the NLS-GFP control strain are shown for detected proteins. Proteins pulled down by GFP-D1 and GFP-Prod ($\log_2FC>1$, $p < 0.05$) are highlighted in yellow.
(XLSX)

**S2 Table. List of proteins detected by LC-MS/MS in ovary lysates from GFP-D1, GFP-Prod, and NLS-GFP strains.** Fold change of average MS1 intensity and adjusted *p*-value for the GFP-D1 and GFP-Prod strains relative to the NLS-GFP control strain are shown for detected proteins. Proteins pulled down by GFP-D1 and GFP-Prod ($\log_2FC>1$, $p < 0.05$) are highlighted in yellow.
(XLSX)

**S3 Table. List of proteins detected by LC-MS/MS in GSC-enriched testes lysates from GFP-D1 and NLS-GFP strains.** Fold change of average MS1 intensity and adjusted *p*-value for the GFP-D1 strain relative to the NLS-GFP control strain are shown for detected proteins.
(XLSX)

**S4 Table. ipTM scores of D1 and Prod interactors based on AFM structural models.** ipTM scores from AFM-based structural modelling between D1/Prod and a subset of identified interactors ($\log_2FC>2$, $p < 0.01$) from all tissues.
(XLSX)

**S5 Table. List of proteins associated with DNA repair in *Drosophila*.** A list of DNA repair proteins in *Drosophila* (GO:0006281) and whether they were identified as a D1/Prod interactor ($\log_2FC>1$, $p < 0.05$) across all tissues.
(XLSX)

**S6 Table. List of proteins associated with transposon repression in *Drosophila*.** A list of transposon repression proteins identified from genome-wide screens in *Drosophila* [71,72] and whether they were identified as a D1/Prod interactor ($\log_2FC>1$, $p < 0.05$) across all tissues.
(XLSX)

**S7 Table. List of proteins detected by LC-MS/MS in embryonic lysates from GFP-Piwi and NLS-GFP strains.** Fold change of average MS1 intensity and adjusted *p*-value for the GFP-Piwi strain relative to the NLS-GFP control strain is shown for detected proteins.
(XLSX)

**S8 Table. ipTM scores of protein pairs based on AFM structural models.** ipTM scores from AFM-based structural modelling of all pairwise interactions between D1, Prod, Piwi, and 4 common interacting proteins (CG14715, CG10208, fit and Ugt35D1).
(XLSX)

**S9 Table. Differentially expressed genes and TE families between *D1* heterozygous and *D1* mutant ovaries in a *chk2^6006* background.** Identified genes were cross-referenced against genes implicated in TE silencing from S6 Table.
(XLSX)

**S1 Raw Images. Uncropped and unadjusted images from western blots associated with S1 and S4 Figs.**
(PDF)

**S1 Data. The number of D1 foci per germline stem cell, Prod foci per germline stem cell as well as the overall male and female fertility are shown for the indicated strains.**
(XLSX)

**S2 Data. A list of proteins found to interact with D1 alone, Prod alone, or both across all tissues.** These values were used to generate the intersection plot.
(XLSX)

**S3 Data. Abundance (rpm) and size profiles of miRNAs and small RNAs and antisense TE-mapping piRNAs from 3 independent replicates of heterozygote control and D1 mutant ovaries.**
(XLSX)

**S4 Data. Raw data from the image analysis of chromocenter size, Cutoff intensity and 359 bp plus strand transcript intensity from heterozygote control and D1 mutant nurse cells.**
(XLSX)

**S5 Data. Raw values of germaria morphology, fertility, and *copia* expression in germline stem cells from the indicated genotypes.**
(XLSX)

**S6 Data. Raw values indicating percentage of progeny per cross that were homozygous for the *prod^k* loss-of-function allele.**
(XLSX)

**S7 Data. The number of D1 foci and Prod foci per germline stem cell from the indicated genotypes.**
(XLSX)

## Acknowledgments

We thank members of the Jagannathan lab, Ulrike Kutay, Ruth Kroschewski, and Hugo Stocker for discussion and comments on the manuscript. We thank Benjamin Frühbauer for help with representing the protein models. We thank Daniel Barbash, Ferenc Jankovics, Yukiko Yamashita, the Bloomington Drosophila Stock Center, the Vienna Drosophila

Resource Center, the Kyoto Drosophila Stock Center and the Developmental Studies Hybrid-oma Bank for reagents and resources. We thank the Scientific Center for optical and Electron Microscopy (ScopeM) and the Mass Spectrometry facility at the Institute of Biochemistry at ETH Zurich for technical support. We thank Shinichi Sunagawa for providing bioinformatics resources. AC acknowledges support from Genetics Society of America in the form of a DeLill Nasser Award for Professional Development in Genetics.

## Author Contributions

**Conceptualization:** Madhav Jagannathan.

**Formal analysis:** Laszlo Tirian, Dominik Handler, Anna Sintsova.

**Funding acquisition:** Madhav Jagannathan.

**Investigation:** Ankita Chavan, Lena Skrutl, Federico Uliana, Melanie Pfister, Franziska Brändle, Laszlo Tirian, Delora Baptista, David Burke, Anna Sintsova, Madhav Jagannathan.

**Methodology:** Ankita Chavan, Lena Skrutl, Federico Uliana, Melanie Pfister, Franziska Brändle, Laszlo Tirian, Delora Baptista, David Burke, Madhav Jagannathan.

**Supervision:** Madhav Jagannathan.

**Writing – original draft:** Ankita Chavan, Lena Skrutl, Federico Uliana, David Burke, Madhav Jagannathan.

**Writing – review & editing:** Ankita Chavan, Lena Skrutl, Federico Uliana, Franziska Brändle, Anna Sintsova, Pedro Beltrao, Julius Brennecke, Madhav Jagannathan.

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
