## [Editor Report · Decision Letter 0]

25 Sep 2024

Dear Dr Jagannathan, 

Thank you for submitting your revised manuscript from Review Commons entitled "Multi-tissue proteomics identifies a link between satellite DNA organization and heritable transposon repression in Drosophila" for consideration as a Research Article by PLOS Biology. Please accept my sincere apologies for the delay in getting back to you with feedback as we consulted with an academic editor about your submission. 

Your manuscript has now been evaluated by the PLOS Biology editorial staff, as well as by an academic editor with relevant expertise, and I am writing to let you know that we would like to send your submission out for external peer review by the original reviewers at Review Commons. 

IMPORTANT: During our discussions with the Academic Editor, he/she raised some concerns with the statistical tests used for the analysis of the mass spectrometry data since candidates are selected using uncorrected p-values. I have pasted some specific comments from the Academic Editor about this below my signature (labelled 'Comments from the Academic Editor'). I note that Reviewer #2 also raised similar concerns in their review. Therefore, before we send your manuscript back to the previous reviewers, we would like to ask that a change to reporting corrected p-values is made in the manuscript. Please do let us know if this is likely to take a bit of time, as we can close your submission and then consider the revised version as a new submission. 

In addition, after discussions with the rest of the editorial team, we think your manuscript would be a better fit as Resource Article at the journal given the significant value offered by the proteomics dataset (https://journals.plos.org/plosbiology/s/what-we-publish#loc-methods-and-resources-articles). During resubmission (see details below), we would be grateful if you could please 'Methods and Resources' as the article type in the dropdown menu in the online submission form. 

Before we can send your manuscript to reviewers, we need you to complete your submission by providing the metadata that is required for full assessment. To this end, please login to Editorial Manager where you will find the paper in the 'Submissions Needing Revisions' folder on your homepage. Please click 'Revise Submission' from the Action Links and complete all additional questions in the submission questionnaire.

Once your full submission is complete, your paper will undergo a series of checks in preparation for peer review. After your manuscript has passed the checks it will be sent out for review. To provide the metadata for your submission, please Login to Editorial Manager (https://www.editorialmanager.com/pbiology) within two working days, i.e. by Sep 27 2024 11:59PM.

Kind regards,

Richard

Richard Hodge, PhD

rhodge@plos.org

COMMENTS FROM THE ACADEMIC EDITOR 

One thing that concerned me was the use of statistics for the mass spectrometry, particularly the decision to select candidates based on p-value but using the uncorrected p-value rather than the corrected (multiple test adjusted) p-value. This approach is not valid. I think it would be better for the authors to select candidates based on a non-p-value based method, ie semi quantitatively, based on low variability and high fold change (which is effectively what p-value seeks to measure) but admitting that due to the noise in the data they have insufficient power to call p-values.

---

## [Decision Letter · Decision Letter 1]

28 Nov 2024

Dear Madhav,

Thank you for your continued patience while we considered your revised manuscript "Multi-tissue proteomics identifies a link between satellite DNA organization and heritable transposon repression in Drosophila" for consideration as a Methods and Resources article at PLOS Biology. Please accept my sincere apologies for the delays that you have experienced during this round of the peer review process Your revised study has now been evaluated by the PLOS Biology editors, the Academic Editor and the original reviewers at Review Commons. 

In light of the reviews, which you will find at the end of this email, we are pleased to offer you the opportunity to address the remaining points from Reviewer #2 in a revision that we anticipate should not take you very long. We will then assess your revised manuscript and your response to the reviewers' comments with our Academic Editor. 

In addition, I would be grateful if you could please address the following editorial and data-related requests that I have provided below (A-H):

(A) We routinely suggest changes to titles to ensure maximum accessibility for a broad, non-specialist readership. In this case, we would suggest a minor edit to the title as follows, to highlight the resource value of the manuscript. Please ensure you change both the manuscript file and the online submission system, as they need to match for final acceptance:

“Proteomic analyses reveal the constitutive heterochromatin-associated proteome in multiple Drosophila tissues”

(B) You may be aware of the PLOS Data Policy, which requires that all data be made available without restriction: http://journals.plos.org/plosbiology/s/data-availability. For more information, please also see this editorial: http://dx.doi.org/10.1371/journal.pbio.1001797

-Supplementary files (e.g., excel). Please ensure that all data files are uploaded as 'Supporting Information' and are invariably referred to (in the manuscript, figure legends, and the Description field when uploading your files) using the following format verbatim: S1 Data, S2 Data, etc. Multiple panels of a single or even several figures can be included as multiple sheets in one excel file that is saved using exactly the following convention: S1_Data.xlsx (using an underscore).

-Deposition in a publicly available repository. Please also provide the accession code or a reviewer link so that we may view your data before publication. 

Figure 1C-F, 2A-E, 3C-D, 4B-F, 5C-D, 5G, 6C-D, 6F-G, 6I, S1B, S2A, S5B-C

(C) Please deposit the RNA-seq and small RNAseq data in a public data repository such as the GEO. Please ensure that the data is made publicly available and provide the accession number in the Data Availability Statement in the online submission form. 

(D) Thank you for already depositing the proteomic datasets in the ProteomeXchange. I have tried to search the accession numbers provided but it seems that the data is not yet publicly available. I would be grateful if you could please make the data publicly available at this stage before publication.

(E) Please also ensure that each of the relevant figure legends in your manuscript include information on *WHERE THE UNDERLYING DATA CAN BE FOUND*, and ensure your supplemental data file/s has a legend.

(F) We require the original, uncropped and minimally adjusted images supporting all blot and gel results reported in the following Figures:

Figure S1D-E, S4B

We will require these files before a manuscript can be accepted so please prepare and upload them now. Please carefully read our guidelines for how to prepare and upload this data: https://journals.plos.org/plosbiology/s/figures#loc-blot-and-gel-reporting-requirements

(G) Please ensure that your Data Statement in the submission system accurately describes where your data can be found and is in final format, as it will be published as written there. 

(H) Per journal policy, if you have generated any custom code during the course of this investigation, please make it available without restrictions. Please ensure that the code is sufficiently well documented and reusable, and that your Data Statement in the Editorial Manager submission system accurately describes where your code can be found. 

**IMPORTANT - SUBMITTING YOUR REVISION**

*Resubmission Checklist*

*Published Peer Review*

*PLOS Data Policy*

*Blot and Gel Data Policy*

Kind regards,

Richard

Richard Hodge, PhD

rhodge@plos.org

REVIEWS:

Reviewer #1: Comments to Chavan et al.

The authors addressed all my comments in a satisfying manner. The proteomic data are now much better described and the new RNA-Seq data clearly distinguish the effects of D1 from the ones of the RDC complex. The authors also made several editorial changes making the manuscript much easier to read.

Reviewer #2: In this revised manuscript, Chavan et al. have substantially improved the clarity, rigor, and depth of their report by revisions to the text and the addition of many new experiments, including additional QC on the proteomics experiments, RNA-seq/piRNA-seq experiments, and a deeper analysis of the D1 mutants. 

In short, the proteomics data offer a unique resource to the community while the more focused analysis linking the piRNA pathway and D1 illuminates new players in small RNA biogenesis and TE repression in an already well-studied system. Please see below for my major and minor comments on this new version of the manuscript:

Major comments:

1. I was surprised to learn that the single IP-WB could not validate the Piwi:D1 interaction. The explanation of the interaction occurring in only a few cells makes sense given that the authors had to wait until the F2 generation to detect a defect; however, why would such a rare interaction in the larger embryo (or a transient one, as was also suggested) be detect robustly by the AP-MS? If this is simply a detection issue, the authors should assert that for the uninitiated. 

2. I'm afraid the original TE data, the figures of which have been shunted to the supplement, do not add to the main text. I'm aware that this was a ton of work but it's distracting now that the authors have supplied RNA-seq data. 

3. The authors mention that tirant was a rare TE upregulated in the D1 mutant ovaries; however, the piRNA data suggest that this upregulation is not the production of piRNA pathway perturbations. This inconsistency should be addressed. Is tirant upregulation independent of the piRNA pathway? Related: the authors might consider inverting the presentation to show the piRNA-seq first and RNA-seq second given the a priori hypothesis that that the piRNA pathway should be disrupted and TE repression would the product of that. 

4. The most dramatic phenotype is the ovary atrophy in the F2 ovaries. However, these were not subjected to RNA-seq and instead a (very convincing) copia RNA FISH experiment. There was no clear justification for 1. Focusing on copia (is copia a particularly excellent TE for learning about a larger disruption to TE silencing?). And 2. Why assaying copia piRNAs is not necessary for implicating the piRNA pathway. If these cannot be addressed in the text, an RNA-seq experiment (and possibly a piRNA-seq experiment) should be performed. 

Minor comments:

5. More cautious language should be used around the link between TE depression and Chk2 rescue. The authors detect many DNA repair proteins interacting directly or indirectly with D1. The disruption to DNA repair at satellite-rich DNA could account for the DNA damage as well, no?

6. Abstract Line 40/41: transcriptomics and small RNA profiling was not used for the F2 phenotypic analysis - only RNA FISH was, correct?

7. Figure 2A: Are there no proteins that both repress TEs and were common to the two experiments, as the color coding suggests (there is no category for both)?

8. Lines 147-155: Observing these positive controls in the dataset is certainly satisfying. However, the authors failed to mention any known interactors that did not show up in their new dataset.

9. Fig 2C: Is LogFC >1 an agreed upon cut-off for this type of data? A LogFC of 1.1 is not especially compelling to the outside reader. Similar, in Fig 2D: why are proteins displayed that have a p[adj] >0.1?

10. Fig 3A: showing "co-localization" with a protein that diffusely marks the whole nucleus seems rather uninformative; moreover, the absence of single channels would make it difficult to see overlap if indeed it was informative.

11. Fig 5: more striking than the elevated 359bp signal is probably the shift from distinct foci to a single, large blob-like signal. Given that D1 is a sort of architectural protein, this interesting finding requires at least some mention.

12. The RNAaseH experiment lacks a positive control without the treatment.

Reviewer #3 (Fu Yang, signs review): The authors have addressed all my questions, and I have no further questions.

---

## [Editor Report · Decision Letter 2]

12 Dec 2024

Dear Madhav,

On behalf of my colleagues and the Academic Editor, Peter Sarkies, I am pleased to say that we can accept your manuscript for publication, provided you address any remaining formatting and reporting issues. These will be detailed in an email you should receive within 2-3 business days from our colleagues in the journal operations team; no action is required from you until then. Please note that we will not be able to formally accept your manuscript and schedule it for publication until you have completed any requested changes.

PRESS

Best wishes, 

Richard

Richard Hodge, PhD

rhodge@plos.org

PLOS
